# Embryonic and postnatal neurogenesis produce functionally distinct subclasses of dopaminergic neuron

Elisa Galliano[1,2,3]*, Eleonora Franzoni[1], Marine Breton[1†], Annisa N Chand[1‡], Darren J Byrne[1], Venkatesh N Murthy[2,3], Matthew S Grubb[1]*

[1]Centre for Developmental Neurobiology, Institute of Psychiatry, Psychology and Neuroscience, King's College London, London, United Kingdom; [2]Department of Molecular and Cellular Biology, Harvard University, Cambridge, United States; [3]Centre for Brain Science , Harvard University, Cambridge, United States

**Abstract** Most neurogenesis in the mammalian brain is completed embryonically, but in certain areas the production of neurons continues throughout postnatal life. The functional properties of mature postnatally generated neurons often match those of their embryonically produced counterparts. However, we show here that in the olfactory bulb (OB), embryonic and postnatal neurogenesis produce functionally distinct subpopulations of dopaminergic (DA) neurons. We define two subclasses of OB DA neuron by the presence or absence of a key subcellular specialisation: the axon initial segment (AIS). Large AIS-positive axon-bearing DA neurons are exclusively produced during early embryonic stages, leaving small anaxonic AIS-negative cells as the only DA subtype generated via adult neurogenesis. These populations are functionally distinct: large DA cells are more excitable, yet display weaker and – for certain long-latency or inhibitory events – more broadly tuned responses to odorant stimuli. Embryonic and postnatal neurogenesis can therefore generate distinct neuronal subclasses, placing important constraints on the functional roles of adult-born neurons in sensory processing.

DOI: https://doi.org/10.7554/eLife.32373.001

*For correspondence: elisa.galliano@kcl.ac.uk (EG); matthew.grubb@kcl.ac.uk (MSG)

Present address: [†]Centre de Recherche en Neurosciences de Lyon, Inserm U1028, Lyon, France; [‡]Institut für Neuropathologie, Charité - Universitätsmedizin Berlin, Berlin, Germany

Competing interests: The authors declare that no competing interests exist.

## Introduction

The adult central nervous system has been long believed to be incapable of self-regeneration. Five decades ago, however, pioneering studies revealed the existence of well-defined neurogenic niches in the brain of adult rodents (*Altman, 1962*). Such neurogenic zones are small, spatially defined and give rise to two broad neuronal populations: hippocampal dentate granule cells, and interneurons in the olfactory bulb (OB) (*Lledo et al., 2006*). These newly generated cells are believed to bring unique properties to existing networks, largely by virtue of the specialised functional and plastic features associated with their transient immature status (*e.g. Carleton et al., 2003*; *Ge et al., 2007*; *Gu et al., 2012*; *Livneh et al., 2014*; *Marín-Burgin and Schinder, 2012*; *Nissant et al., 2009*; *Schmidt-Hieber et al., 2004*, but see *Sailor et al., 2016*). Once fully mature, though, the functional properties of adult-generated neurons in both the hippocampus and olfactory bulb often closely match those of their developmentally generated neighbours (*e.g. Carleton et al., 2003*; *Grubb et al., 2008*; *Laplagne et al., 2006*; *Marín-Burgin and Schinder, 2012*; *Nissant et al., 2009*, but see *Valley et al., 2013*). Does this mean, then, that within broad classes of neuron – for example, dentate granule cells, or OB granule cells – embryonic and postnatal neurogenesis always produce fundamentally similar cell types?

Among the wider population of adult-generated OB cells is a heterogeneous group of inhibitory neurons situated in the structure's glomerular layer, whose main role is to modulate the earliest

**eLife digest** Most of your brain cells were born before you were. But in mammals, including humans, some of these brain cells, also known as nerve cells or neurons, are created after birth. These later-generated neurons are often extremely similar to their counterparts produced in the womb, and also seem to perform a similar role once they are fully mature. However, it has not been entirely clear if the later-produced neurons may also have a specific purpose.

Neurons are made of a cell body with a cable-like structure called axon that transmits information to more distant neurons, and dendrites, which are branches that receive information from other neurons. Neurons use different signalling molecules to communicate, one of which is called dopamine, and the neurons that use this specific signal are called dopaminergic neurons.

Now, Galliano et al. wanted to test if neurons created in the womb, and neurons created after birth, are really so similar. To investigate this, they compared the dopaminergic neurons from mice found in the first part of the brain to process information about smell – the olfactory bulb. These specific neurons are known to have diverse properties and can also be produced after birth.

Galliano et al. studied their development, form and purpose, and discovered that only neurons produced in the womb can possess an axon. Moreover, the axon-bearing cells had a different form and functional properties to their axon-less cousins, and also showed some subtle differences in their ability to respond to smell. This demonstrates that two very different types of dopaminergic neurons in the olfactory bulb are produced at different stages during the development.

A better knowledge of such basic brain-developmental features is essential for the wider goal of understanding how the brain operates, and to discover ways to repair it when it is not working properly. Neurons created after birth in particular, might enable us to develop new treatment strategies; for example, adding new dopaminergic neurons to replace those lost in degenerative disorders such as Parkinson's Disease. When developing such regenerative therapies, why not learn lessons from how the brain can achieve this naturally?

DOI: https://doi.org/10.7554/eLife.32373.002

stages of sensory information processing (*Alonso et al., 2012*; *Fukunaga et al., 2014*; *Grubb et al., 2008*; *Livneh et al., 2014*; *Livneh and Mizrahi, 2012*). Different subclasses of glomerular layer interneuron can be identified by their specific expression of calcium-binding proteins (*Kosaka and Kosaka, 2011*), while another major subclass is identified by its ability to co-release both GABA and dopamine (*Borisovska et al., 2013*; *Vaaga et al., 2017*). These dopaminergic (DA) interneurons can be generated via adult neurogenesis (*Adam and Mizrahi, 2011*; *Bonzano et al., 2016*; *De Marchis et al., 2007*), and the survival of postnatally generated DA cells is activity-dependent (*Bonzano et al., 2014*). Moreover, adult-born DA cells have been shown to contribute to specific olfactory behaviours (*Lazarini et al., 2014*).

In recent years, the manner in which resident and adult-generated DA neurons contribute to olfactory processing has been widely studied, and there is now an accumulating - and sometimes contrasting - body of evidence on the role played by DA cells in glomerular circuits. A wide spectrum of functions has been proposed for these cells, involving either local or broadly distributed actions within the glomerular layer, and roles as diverse as modulating release from olfactory sensory neuron terminals, signal normalisation, contrast enhancement, and temporal decorrelation (*Banerjee et al., 2015*; *Cavarretta et al., 2016*; *Economo et al., 2016*; *Liu et al., 2013*; *Mainland et al., 2014*; *Pignatelli and Belluzzi, 2017*; *Roland et al., 2016*; *Vaaga et al., 2017*). The complexity and sometimes mutually exclusive nature of such functions – especially in relation to spatial connectivity – make it unlikely that a single class of interneuron could perform them all. Indeed, morphological variability has been demonstrated among OB DA neurons and has been linked to their time of birth (*Halász et al., 1981*; *Kiyokage et al., 2010*; *Kosaka and Kosaka, 2007*; *Kosaka et al., 2008*; *Kosaka and Kosaka, 2009*; *Kosaka and Kosaka, 2011*; *Kosaka and Kosaka, 2016*; *McLean and Shipley, 1988*; *Pignatelli and Belluzzi, 2017*; *Pignatelli et al., 2005*). However, no discrete features demarcating distinct OB DA subpopulations have yet been identified. More importantly, nothing is currently known regarding the functional properties of putative OB DA subtypes. Are there physiological differences between embryonically generated and adult-born DA cells? And might such

differences start to account for the various functional roles ascribed to this cell type in sensory processing?

Here, we build on previous work *in vitro* (*Chand et al., 2015*), to show that different classes of OB DA neuron *in vivo* can be clearly distinguished based on the presence or absence of an axon and its key subcellular specialisation, the axon initial segment (AIS). AIS-positive DA cells are larger, with broader dendritic arborisations, and are exclusively born in early embryonic development. Postnatally generated DA cells, in contrast, are all small and anaxonic. Crucially, these morphological and ontological distinctions also map onto clear functional differences in both cellular excitability and odorant response properties *in vivo*, strongly constraining the potential role of adult-born DA cells in sensory processing.

## Results

### The axon initial segment is only present in a distinct subset of DA cell

To investigate the presence of an axon initial segment (AIS) in DA cells we performed immunohistochemistry on fixed slices of the olfactory bulb of juvenile (P28) wild-type C57/Bl6 mice. We identified DA cells by labelling them with an antibody against tyrosine hydroxylase (TH), the rate-limiting enzyme in the biosynthesis of dopamine. For AIS identification we stained for ankyrin-G, the master AIS organising molecule (AnkG, *Figure 1A*) (*Hedstrom et al., 2008*; *Jenkins et al., 2015*; *Zhou et al., 1998*). While for most TH-positive cells we could not detect AnkG label on any of their processes, we identified a subset of large DA neurons that possessed a clear AnkG-positive AIS (*Figure 1A*, middle and right panel; *Kosaka et al., 2008*). Like their midbrain DA (*González-Cabrera et al., 2017*; *Meza et al., 2018*) and cultured (*Chand et al., 2015*) counterparts, the AISs of OB DA neurons were often thin, short and located rather distally away from the soma. Along with the dense and highly interwoven meshwork of TH-positive processes in the glomerular layer (GL), this made AIS-positive cells difficult to identify. Nevertheless, a lower-bound estimate (see Materials and methods) suggests they comprise at least 2.5% of the overall OB DA population.

Soma size quantification revealed these AIS-containing DA neurons to be morphologically distinct. As shown in *Figure 1B*, the soma area distribution of the general bulbar TH+ population is clearly not unimodal (blue distribution): most DA cells are relatively small (peak 55 $\mu m^2$), but there is a distinct minority that are significantly larger (peak 140 $\mu m^2$) (*Kosaka and Kosaka, 2007*; *McLean and Shipley, 1988*; *Pignatelli et al., 2005*). In contrast, performing a similar analysis solely for TH+/AnkG+ cells (*i.e.* DA neurons with an AIS) produced a unimodal distribution centred on the large-cell peak of the full population curve (*Figure 1B*, magenta line; peak 137 $\mu m^2$). Large AIS-positive cells therefore represent a distinct sub-population of OB DA neurons.

These large, AIS-positive DA neurons are also located in a specific sub-region of the GL. Dividing the GL into four sub-laminae (*Figure 1A*; see Materials and methods) revealed the overall TH-positive population to be concentrated in the mid-GL (*Figure 1C*). AIS-positive DA neurons, however, were mostly found in the lower portions of the GL towards the external plexiform layer (EPL) border, with very little presence in the upper or mid-GL (*Figure 1C*; *Liberia et al., 2012*); effect of sub-lamina ×cell type in two-way repeated-measures ANOVA, $F_{3,66} = 35.47$, p<0.0001; post-hoc Sidak's test between cell types, upper-GL, p=0.014; mid-GL, p<0.0001; lower-GL, p<0.0001; EPL border, p=0.98; n = 24 slices from N = 3 mice).

### AIS-lacking DA neurons are anaxonic

The AIS is crucial for the maintenance of axo-dendritic neuronal polarity (*Hedstrom et al., 2008*), and is often employed as an indicator of axonal identity (e.g. *Watanabe et al., 2012*), so does the absence of an AIS in the majority of small DA neurons mean that these cells do not possess an axon? Addressing this question required us to be able to identify and follow *all* of a given cell's individual processes. We therefore achieved sparse label of individual OB DA neurons, either by injecting floxed GFP-encoding viruses (either AAV or RV::dio) in embryos or neonates from VGAT-[Cre] or DAT-[Cre] reporter lines, or by electroporating GFP-encoding plasmid DNA in wild-type neonates (see Materials and methods). The dopaminergic phenotype of the infected neurons was confirmed by immunohistochemical label for TH. We then adopted a dual strategy for axon identification.

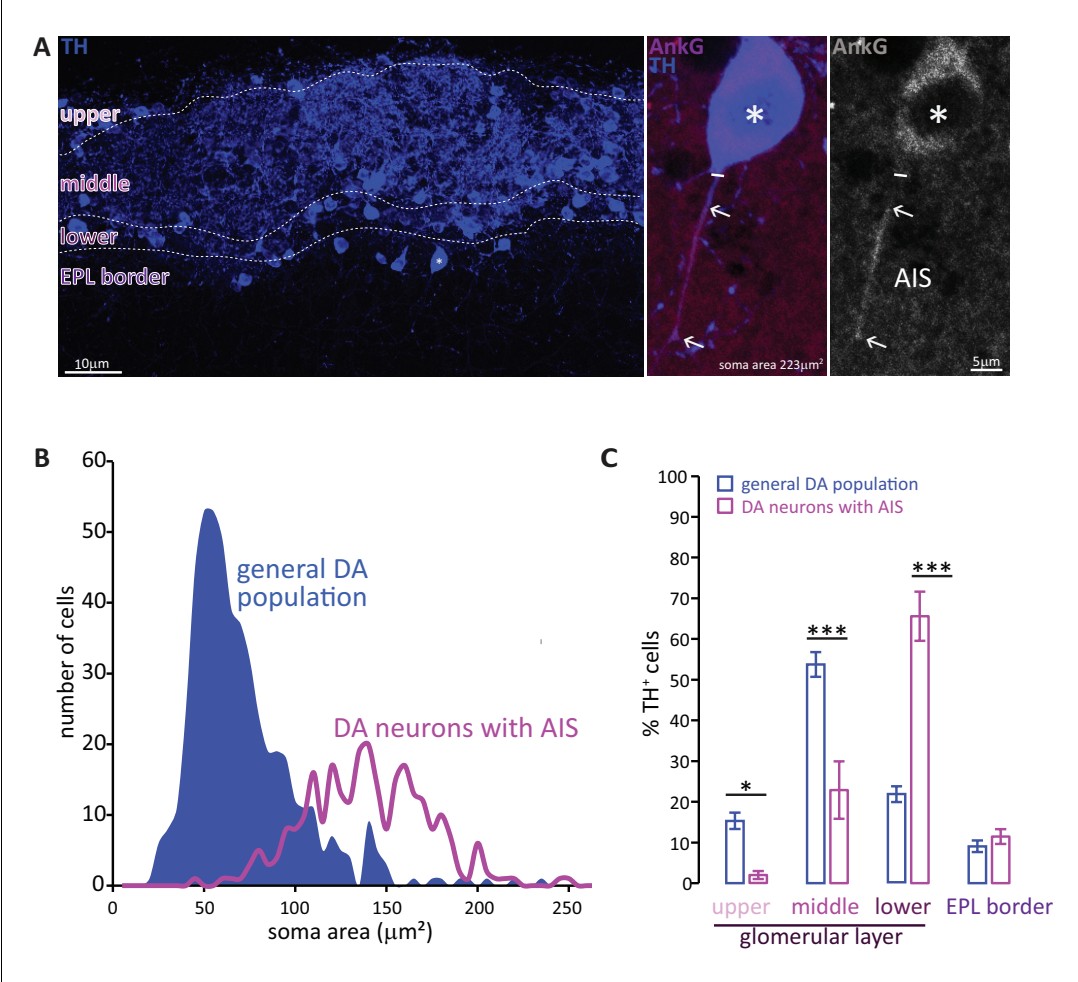

**Figure 1.** Two subtypes of DA neuron can be characterised based on size, location and presence of an AIS. (**A**) Left: example image of olfactory bulb stained with an anti-TH antibody (blue). Dashed lines indicate subregions of the glomerular layer (GL). The asterisk indicates an AIS-positive DA cell. Middle, right: zoomed image of the asterisked cell from the left panel, co-stained for TH (blue) and the AIS marker ankyrin-G (AnkG, magenta or greyscale). The solid line indicates the emergence of the axonal process from the soma; arrows indicate AIS start and end positions. (**B**) Frequency plots showing soma area of the general DA population (blue, filled area, n = 519, N = 3), and of the subset of DA neurons that possesses an AIS (magenta line, n = 271, N = 6). (**C**) Locations in the GL for both the general DA population (blue, mean ± sem; n = 888, N = 6) and AIS-positive DA neurons (magenta, mean ± sem; n = 127, N = 6). Post-hoc Sidak's test following two-way repeated measures ANOVA; *p<0.05; ***p<0.001.
DOI: https://doi.org/10.7554/eLife.32373.003

First – as a positive control – we confirmed that while the AnkG-positive processes of large AIS-containing DA cells co-localised with the axonal marker TRIM-46 (*Figure 2A;van Beuningen et al., 2015*), this axonal marker was entirely absent from the processes of small OB DA neurons (*Figure 2B*; n = 10, N = 3, average soma area 58 µm²). Second – as a negative control – we analysed the expression of the dendritic marker MAP-2 (*Kosik and Finch, 1987*; *Rolls and Jegla, 2015*; *van Beuningen et al., 2015*). DA cells with an AIS express MAP-2 in all processes, even in the proximal axon (*Figure 2C*). However, as reported for other cell types (*Gumy et al., 2017*; *van Beuningen et al., 2015*), this proximal axonal MAP-2 expression fades where AnkG expression begins, and MAP2 is absent from the post-AnkG portion of the axon (*Figure 2C*). Conversely, AIS-negative DA neurons express MAP-2 along the entire length of all their processes (*Figure 2D*; n = 10, N = 3, average soma area 49 µm²). These data strongly suggest that the presence of an AIS is indicative of axonal identity in OB DA cells, and that the small TH-positive neurons that lack an AIS are truly anaxonic.

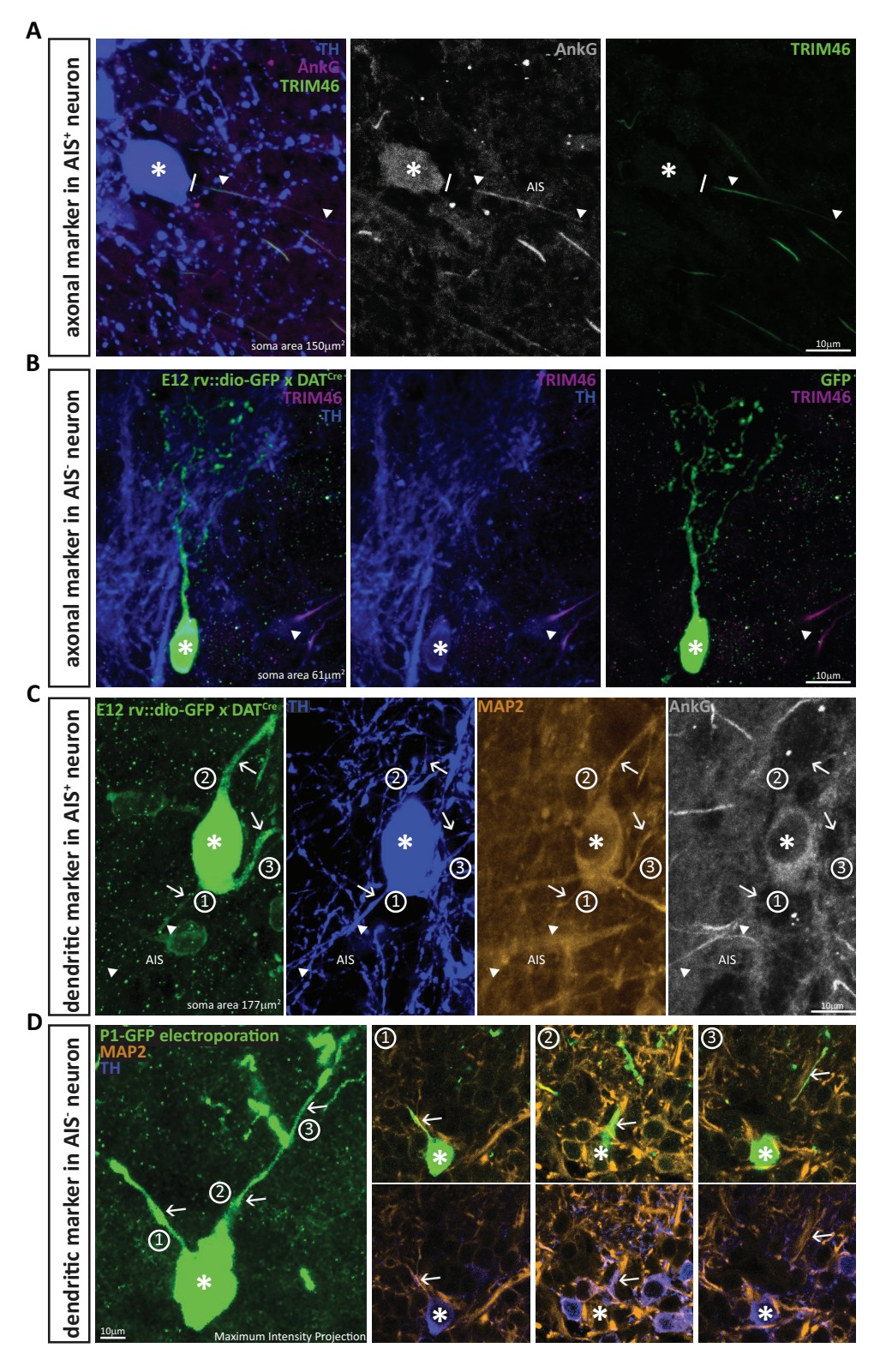

**Figure 2.** DA neurons that lack an AIS also lack the axonal marker TRIM-46, and all their processes co-stain with the dendritic marker MAP-2. (**A**) Example image of a DA cell in a wild-type mouse stained for TH (blue), AnkG (magenta) and the axonal marker TRIM-46 (green). Asterisks indicate soma position; lines indicate the emergence of the axonal process from the soma; triangles indicate AIS start and end positions. (**B**) Example image of an

*Figure 2 continued on next page*

*Figure 2 continued*

anaxonic DA cell in a DAT-Cre mouse injected at E12 with rv::dio-GFP, stained for TH (blue) and TRIM-46 (magenta). Asterisks indicate soma position; triangle shows a TRIM-46- and TH-positive process belonging to a neighbouring, non-GFP-expressing cell. (C) Example image of an AIS-containing DA cell in a DAT-Cre mouse injected at E12 with rv::dio-GFP, stained for TH (blue), MAP-2 (orange) and AnkG (grey). Asterisks indicate soma position; triangles indicate AIS start and end positions; numbers and arrows indicate the three main processes emerging from the soma. Note that MAP-2 fluorescence in the axon (process 1) ends when AnkG fluorescence begins. (D) Left: Maximum intensity projection image of an anaxonic DA cell in a wild-type mouse electroporated with GFP at P1, stained for TH (blue) and MAP-2 (orange). The asterisks indicates soma position; numbers and arrows indicate the three main processes emerging from the soma. Right: panels 1–3 show single z-plane images of each dendritic process, visualised with GFP plus MAP-2 label (top) or TH plus MAP-2 label (bottom). Note that all processes are positive for all three markers.

DOI: https://doi.org/10.7554/eLife.32373.004

## Broader dendritic branching in AIS-positive DA neurons

Sparse labelling of individual OB DA neurons also allowed us to investigate their dendritic morphology (*Figure 3A*), and this again revealed clear differences between AIS-positive and AIS-negative subtypes. Small, anaxonic DA neurons had limited dendritic arborisations that ramified across a small region of the glomerular layer (*Figure 3B,C,E,F*). By contrast, the dendrites of large, axon-bearing DA cells were much more broadly spread (*Figure 3D,E,F*). Despite considerable cell-to-cell morphological variability within each sub-class (*Figure 3E,F*), quantitative Sholl analysis (see Materials and methods) revealed highly significant cell-type differences on multiple dendritic parameters (*Table 1*; *Figure 3E*; effect of cell type in mixed model ANOVA analysis of Sholl distributions, $F_{1,36}$ = 5.30, p=0.027). This is all the more striking given the thin OB slices necessary for AnkG label, and the likely resulting underestimation of glomerular layer ramification by AIS-positive DA neurons.

## AIS-positive DA neurons are exclusively born during early embryonic development, but anaxonic DA cells continue to undergo postnatal and adult neurogenesis

Glomerular layer interneurons in the OB, including TH-positive DA neurons, belong to the highly restricted group of neuronal types capable of regenerating throughout life via adult neurogenesis (*Betarbet et al., 1996*; *Bonzano et al., 2016*; *De Marchis et al., 2007*; *Winner et al., 2002*). This prolonged neurogenic capacity is often considered to be a universal and hallmark feature of these inhibitory interneurons (*e.g. Liu et al., 2013*). However, data from birthdating experiments suggest that – at least on the basis of soma size – OB DA neurons are not homogeneous in their time of generation (*Kosaka and Kosaka, 2009*). This prompted us to ask whether the two morphological subtypes of AIS-positive and AIS-negative OB DA neuron also differ developmentally.

To address this question, we performed classic pulse-chase birthdating experiments. We injected pregnant mice with the thymidine analogue bromodeoxyuridine (BrdU) at different gestational days starting from embryonic day (E) 11, when the nascent olfactory bulb has begun to appear. We then collected tissue from the progeny once they reached one month of age, and from the mothers themselves to analyse adult-generated neurons (*Figure 4A*). We also labelled neonatally generated bulbar interneurons via postnatal electroporation of GFP-encoding plasmid DNA injected into the lateral ventricles at P1 (*Figure 4A*). Cells were then immunostained for BrdU or GFP, along with TH and – because of the histological processing necessary for BrdU detection – a more robust but as yet mysterious marker of the AIS: the unidentified microtubule-associated protein labelled by the 'pIκBα' antibody (*Buffington et al., 2012*). For each injection time point, we analysed all cells that were both BrdU and TH positive, measuring soma area, and noting the presence or absence of an AIS.

The results, presented in *Figure 4B–H*, clearly show that with increasing birth age the soma area distributions lose their right-end tail, indicating that the large DA neurons are mostly born during early development (*Kosaka and Kosaka, 2009*). Interestingly, soma area in adult-born OB DA neurons (*Figure 4H*) appeared small compared to overall DA population distributions at earlier ages, an effect that was not accounted for by either histological processing differences or cell maturation

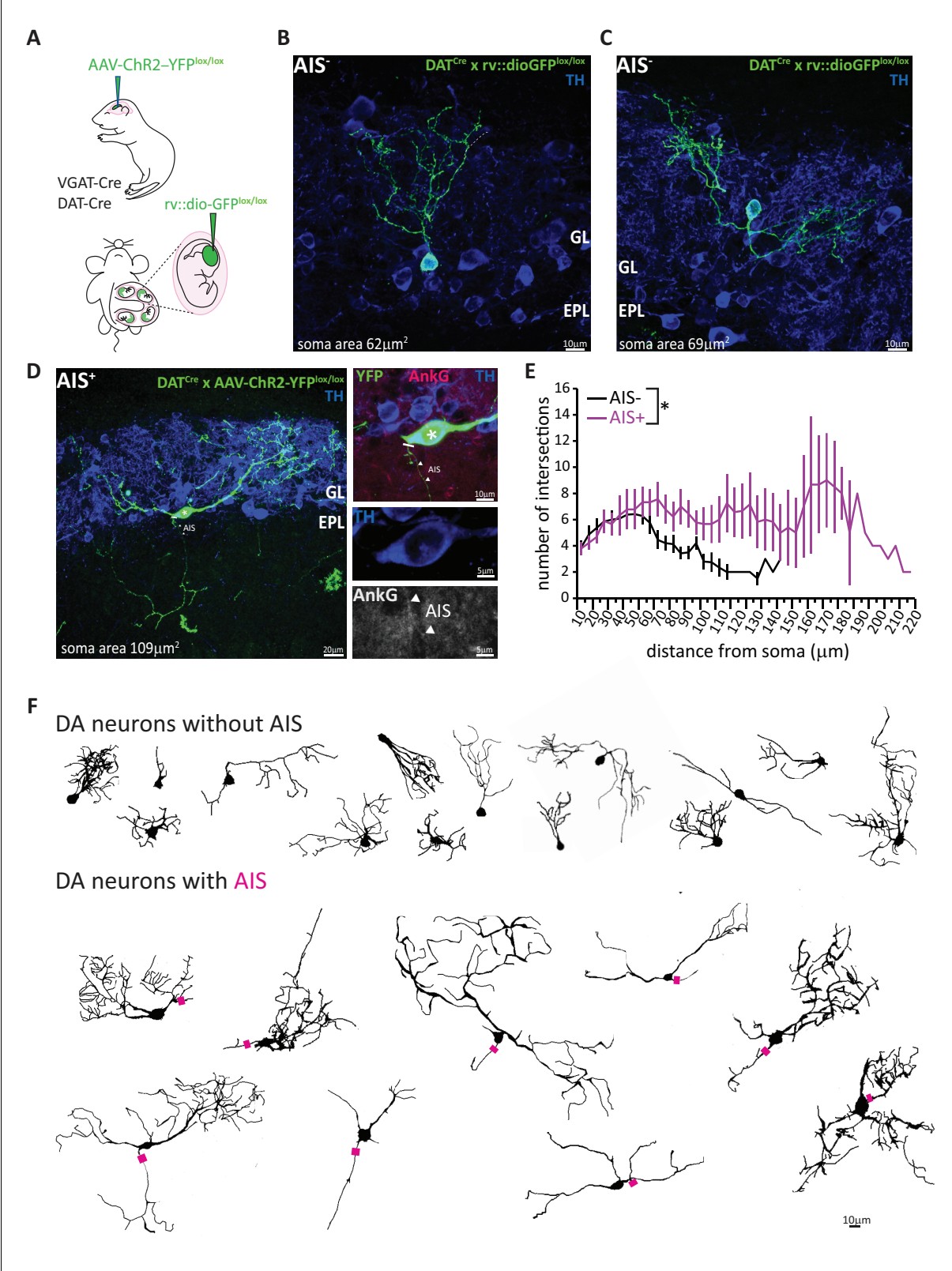

**Figure 3.** AIS-positive DA cells have more widely ramified dendritic morphology. (**A**) Schematic representation of the experimental strategy adopted to achieve sparse labelling of DA cells: P1-2 neonates or E12 embryos from VGAT-Cre or DAT-Cre lines were injected with floxed AAV-YFP or rv:: dio-GFP viruses. Tissue was collected for analysis at P28. (**B–C**) Example images of OB DA cells sparsely labelled with GFP (green), co-stained for TH (blue). GL, glomerular layer; EPL, external plexiform layer. AIS-negative DA cells ramify their dendrites narrowly. (**D**) Example image of a GFP-labelled, AIS-positive

*Figure 3 continued on next page*

*Figure 3 continued*

DA cell which ramifies more broadly, co-stained for TH (blue) and AnkG (magenta). Right: zoomed insets showing GFP, TH and/or AnkG label; line indicates axon start; triangles show AIS start and end positions. (E) Sholl plots of branching patterns for reconstructed DA neurons without (black, n = 14) and with (magenta, n = 9) an AIS. Data points are mean ± SEM; effect of cell type in mixed model ANOVA; *p<0.05; for further quantifications see *Table 1*. (F) Morphological reconstructions of 23 sufficiently sparsely-labelled DA neurons without (top) and with (bottom) an AIS. Approximate AIS location is indicated with a magenta square.

DOI: https://doi.org/10.7554/eLife.32373.005

state, and which might indicate a potential specialization of this DA subpopulation that warrants investigation in future studies.

The developmental distinction between AIS-positive and AIS-negative DA cells, however, was even clearer. AIS-positive DA neurons were exclusively born in embryonic development, with a clear peak in their generation at E11-12 (*Figure 4B,C,H*) and only very few being produced between E13 and E18 (*Figure 4D–F,H*). We did not find a single neonatal- or adult-born OB DA neuron that possessed an AIS (*Figure 4G,H*).

To fully convince ourselves that adult-born DA neurons never possess an AIS, we had to rule out the possibility that these cells took longer than 1.5 months to fully mature. We therefore collected tissue from adult BrdU-injected mice after a prolonged chase period of 4 months (*Figure 5A*). We found very few double-labelled BrdU+/TH +cells (*Figure 5B*; n = 15 cells from N = 2 mice), and all were AIS-negative. The absence of an AIS in adult-generated cells is therefore unlikely due to insufficient maturation time, and instead likely reflects a fundamental characteristic of this OB DA subtype.

These data strongly suggest that prolonged neurogenic capabilities are not a widespread property of bulbar DA neurons. Instead, adult neurogenesis is restricted to the axonless subpopulation, while large AIS-bearing cells are only born during early developmental stages. This finding raised two immediate questions concerning the production and longevity of AIS-positive neurons: 1) is there a preponderance of large AIS-positive DA cells in the neonate? And 2) do embryonically generated AIS-positive neurons persist throughout life? To address the first question, we collected tissue from newborn pups (P0) and quantified the soma area of TH-labelled DA neurons (*Figure 6A*). We indeed found a right-shifted distribution of larger DA neurons at this early postnatal timepoint (*Figure 6B*, Kolmogorov-Smirnov test D = 0.3581, p<0.01; *McLean and Shipley, 1988*). We also obtained a slightly higher lower-bound estimate for the prevalence of AIS-positive DA cells at P0 (*Figure 6C*; see Materials and methods), at ~6% of the overall TH-positive population. In addition, AIS-positive neurons at P0 already have large soma areas (mean ± SEM 98 ± 5 $\mu m^2$, n = 16, N = 2; *Figure 6D*).

**Table 1.** Morphological properties of AIS-negative and AIS-positive DA neurons.

Mean values ± SEM of morphological properties for sparsely labelled AIS-negative (n = 14) and AIS-positive (n = 9) DA cells. Statistical differences between groups (AIS-negative vs AIS-positive) were calculated with a Student's t test for normally distributed data ('t'; with Welch correction '$t^W$') or with a Mann–Whitney test for non-normally distributed data ('MW'). Bold type indicates statistically different measures. Morphological reconstructions and average Sholl plots are presented in *Figure 3*.

**Morphological properties**

| | AIS-negative (mean ± sem) | AIS-positive (mean ± sem) | Test type, *p*-value |
|---|---|---|---|
| Soma area ($\mu m^2$) | 70.49 ± 2.71 | 139.00 ± 17.16 | $t^W$, **0.003** |
| Distance of soma from nerve layer ($\mu m$) | 75.07 ± 12.79 | 141.60 ± 31.38 | MW, **0.02** |
| Number of primary dendrites | 3.14 ± 0.23 | 3.44 ± 0.29 | t, 0.43 |
| Length of primary dendrites ($\mu m$) | 12.14 ± 2.97 | 26.13 ± 5.23 | MW, **0.01** |
| Area under Sholl curve ($\mu m$) | 338.60 ± 42.58 | 871.10 ± 167.20 | $t^W$, **0.01** |
| Furthest intersection ($\mu m$) | 77.86 ± 8.23 | 148.30 ± 13.94 | t, **0.0001** |
| Maximum no. of intersections | 8.36 ± 0.68 | 12.56 ± 1.68 | $t^W$, **0.04** |
| Radius for maximum no. of intersections ($\mu m$) | 36.16 ± 4.45 | 80.00 ± 13.97 | $t^W$, **0.02** |

DOI: https://doi.org/10.7554/eLife.32373.006

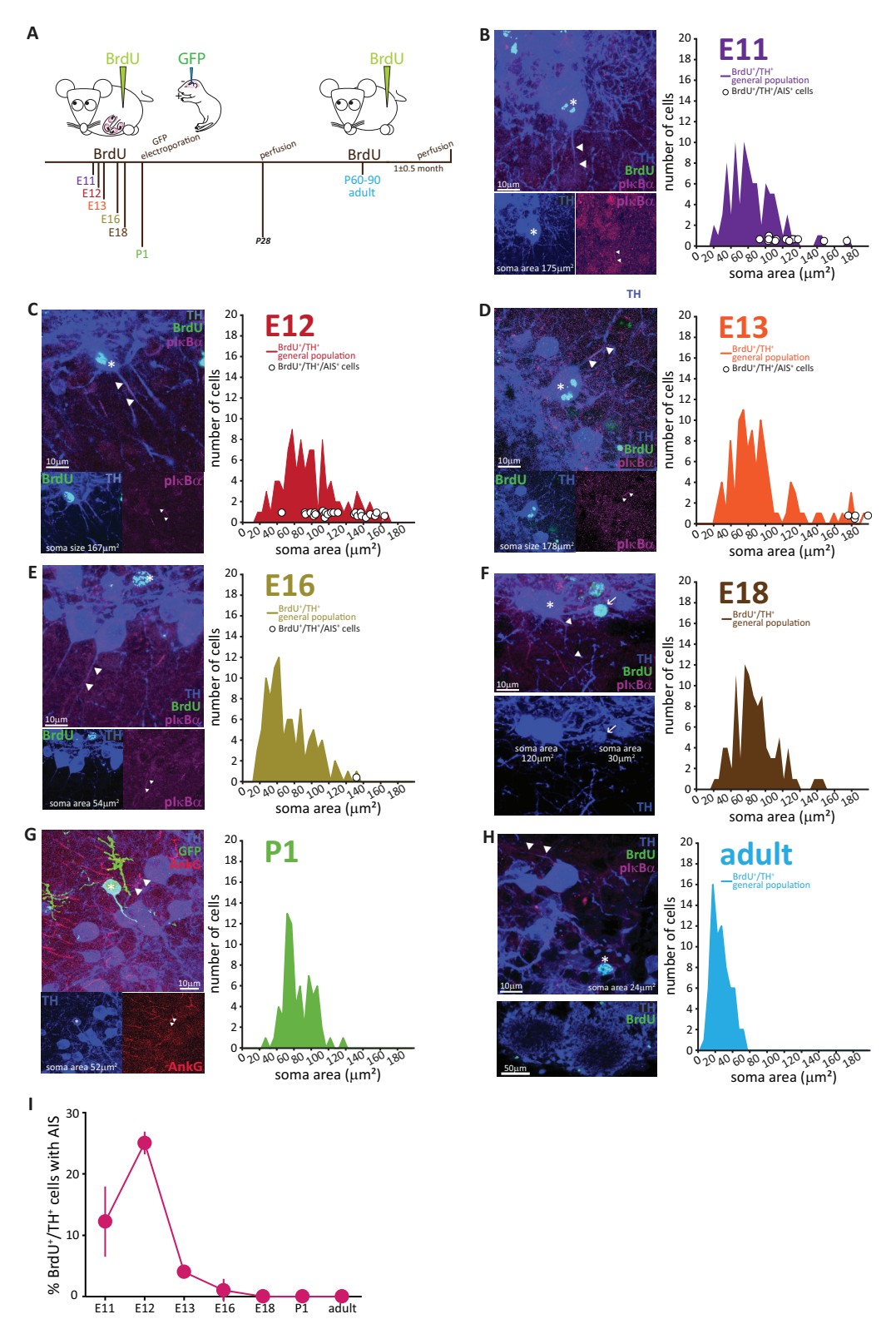

**Figure 4.** AIS-positive DA cells are generated exclusively in early embryonic development. (**A**) Schematic representation of the experimental strategy for birthdating experiments: pregnant wild-type mice were injected with a single dose of BrdU at different gestational days. Tissue was collected from their offspring at 1 month of age (P28), and also from the injected mothers which constitute the 'adult' group. We also labelled neonatally generated cells by electroporating GFP-encoding plasmid DNA injected into the lateral ventricles at P1. (**B–H**) Left: example images from OB slices stained with

*Figure 4 continued on next page*

*Figure 4 continued*

antibodies against BrdU (green), TH (blue) and the AIS markers pIκbα or AnkG (magenta). Asterisks indicate BrdU+/TH +cells; triangles indicate AIS start and end positions; in E-H AISes are indicated in BrDU- cells for comparison. Right: soma area distribution of BrdU+/TH+ DA cells (for each BrdU time point n = 100, N = 3; for P1 electroporation n = 68, N = 3; for adult mothers n = 70, N = 2). Circles indicate AIS-positive cells (BrdU+/TH+/pIκBα+) at their respective soma size value. (I) Summary graph indicating the percentage of AIS-positive cells (BrdU+/TH+/pIκBα+) generated at each developmental time point (mean ± SEM). No AIS-positive DA cells were born after E18.

DOI: https://doi.org/10.7554/eLife.32373.007

To address the second question concerning the longevity of AIS-positive DA neurons, we employed a prolonged pulse-chase birthdating protocol. Mice were injected with BrdU at E12, then left until 6 months of age, when we collected tissue and looked for AIS-positive DA neurons (*Figure 7A*). As shown in *Figure 7B*, we were still able to find DA neurons born at E12 in the bulb of these fully adult mice. All these neurons were large and the overwhelming majority carried an AIS. When we compared these adult animals to littermates that had also been injected at E12 but perfused at P28, we found that E12-born DA neurons in adult animals were larger (91 ± 4 $\mu m^2$; *Figure 7C,D*). Moreover, while in 1-month-old animals TH+/E12-BrdU+ cells were relatively abundant but only 25% of them were AIS-positive, in adult mice TH+/E12-BrdU+ cells were rare but over 90% of them possessed an AIS (*Figure 7E*). These findings strongly suggest that, of the mixed pool of DA neurons born at E12, the initially abundant AIS-negative cells are turned over at some point before 6 months of age, to be substituted by postnatallyborn AIS-negative neurons. In contrast, embryonicallygenerated, large AIS-positive cells can persist throughout adult life.

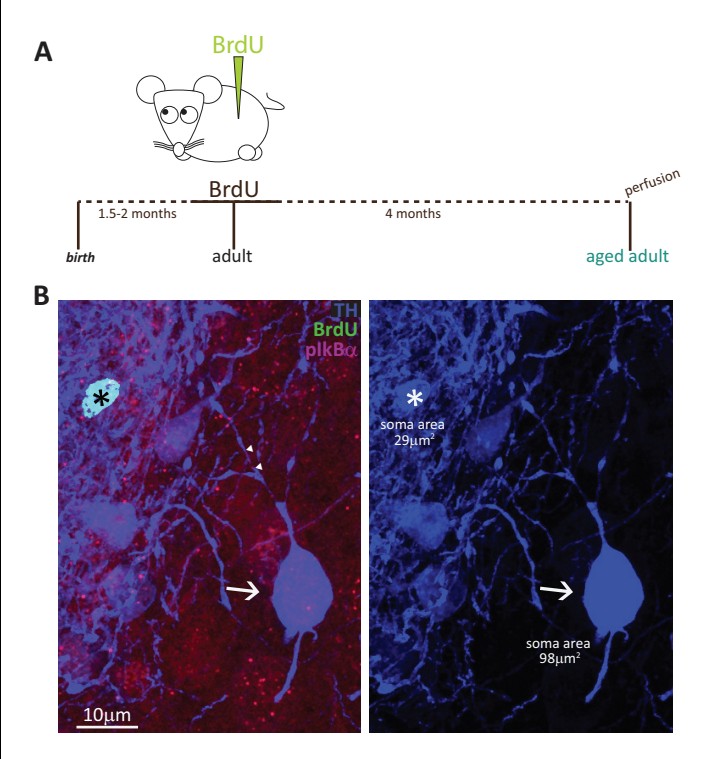

**Figure 5.** Fully mature adult-born DA neurons are small and never possess an AIS. (A) Schematic representation of the experimental strategy: pregnant wild-type mice were injected with a single dose of BrdU, and their tissue was collected 4 months later to allow full maturation of adult-born neurons. (B) Example image of adult tissue perfused 4 months post-BrdU injection, stained with antibodies against BrdU (green), TH (blue) and the AIS marker pIκBα (magenta). Asterisk indicates a BrdU +DA cell (TH+/pIκBα-), triangles indicate AIS start and end positions; the arrow indicates a neighbouring large AIS +DA neuron that is BrdU-negative.
DOI: https://doi.org/10.7554/eLife.32373.008

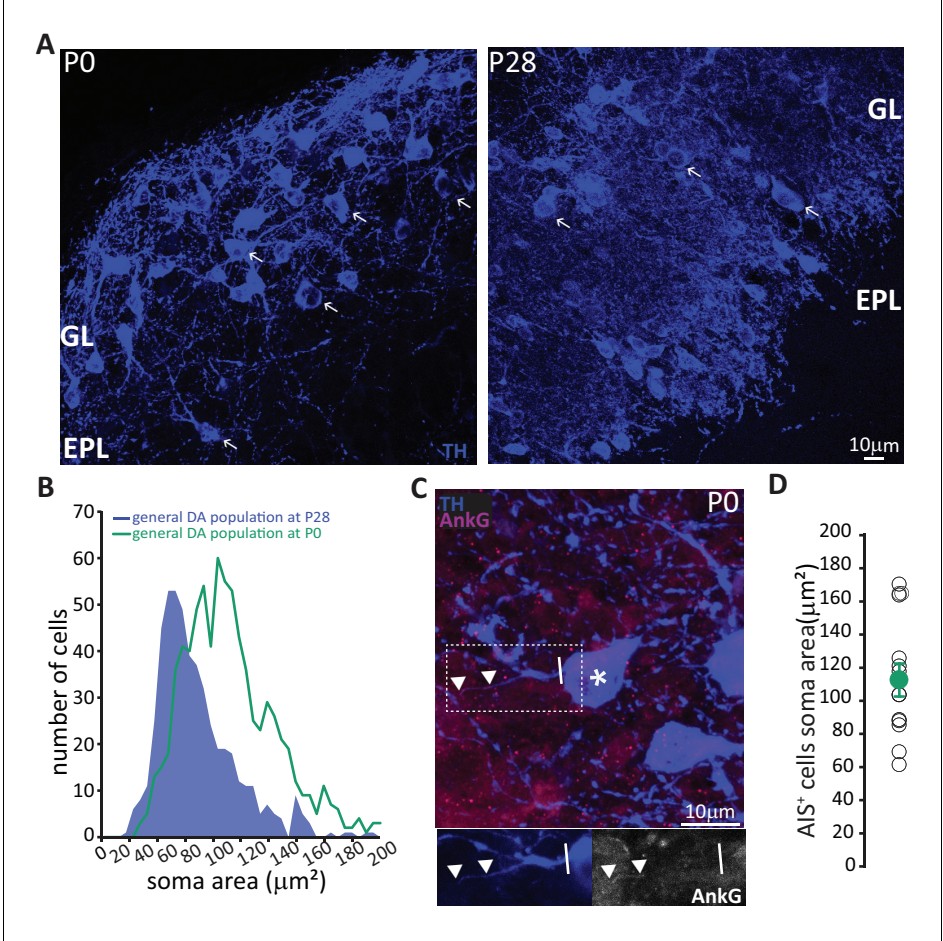

**Figure 6.** In neonates there is a preponderance of large DA neurons that can already possess an AIS. (**A**) Low-magnification example images of tissue from P0 and P28 wild-type mice stained with an antibody against TH (blue). Arrows indicate cells with a soma area bigger than 100 μm²; 'GL' indicates glomerular layer, 'EPL' indicates external plexiform layer. (**B**) Soma area distribution of TH +DA cells in P0 mice (teal, n = 781, N = 3), overlaid on the soma area distribution of the general DA cell population at P28 (blue filled line; see *Figure 1B*); Kolmogorov-Smirnov test between the two distributions **, D = 0.3581, p<0.01. (**C**) High-magnification example image of tissue from a P0 mouse stained with antibodies against TH (blue) and AnkG (magenta). Asterisk indicates the soma of an AIS-positive cell; dashed lines show the inset area magnified below; solid line shows axon start; triangles indicate AIS start and end positions. (**D**) Soma area of TH+/AnkG+ DA cells in P0 mice. Empty circles represent individual cells, full circle shows mean ± SEM (110 ± 10 μm²; n = 16, N = 2).
DOI: https://doi.org/10.7554/eLife.32373.009

## AIS-positive DA neurons possess distinct intrinsic functional properties

Our morphological and developmental analyses revealed a clear distinction between - on the one hand – embryonically born, large and widely branching DA neurons with an axon and AIS, and – on the other – lifelong-generated, small and locally ramifying anaxonic DA cells that do not have an AIS. But do these marked ontological and structural differences also translate into functional heterogeneity? To test this hypothesis, we first performed whole-cell current-clamp electrophysiological recordings on DA neurons in acute ex vivo OB slices.

We visualised DA neurons by crossing the dopaminergic reporter line DAT-Cre (*Bäckman et al., 2006*) with a floxed tdTomato (tdT) reporter line (*Madisen et al., 2010*). In the resulting DAT-tdT mice, the majority of TH+ DA cells also expressed tdT (90 ± 4% of all TH+ neurons were also tdT+; n = 369, N = 3; *Figure 8A*). The rare TH+ neurons that lacked tdT fluorescence tended to be large (*Figure 8B*), suggesting that while our genetic labelling strategy comprehensively identified small, anaxonic DA neurons, it under-represented the large, AIS-positive DA subtype. Indeed, co-label for

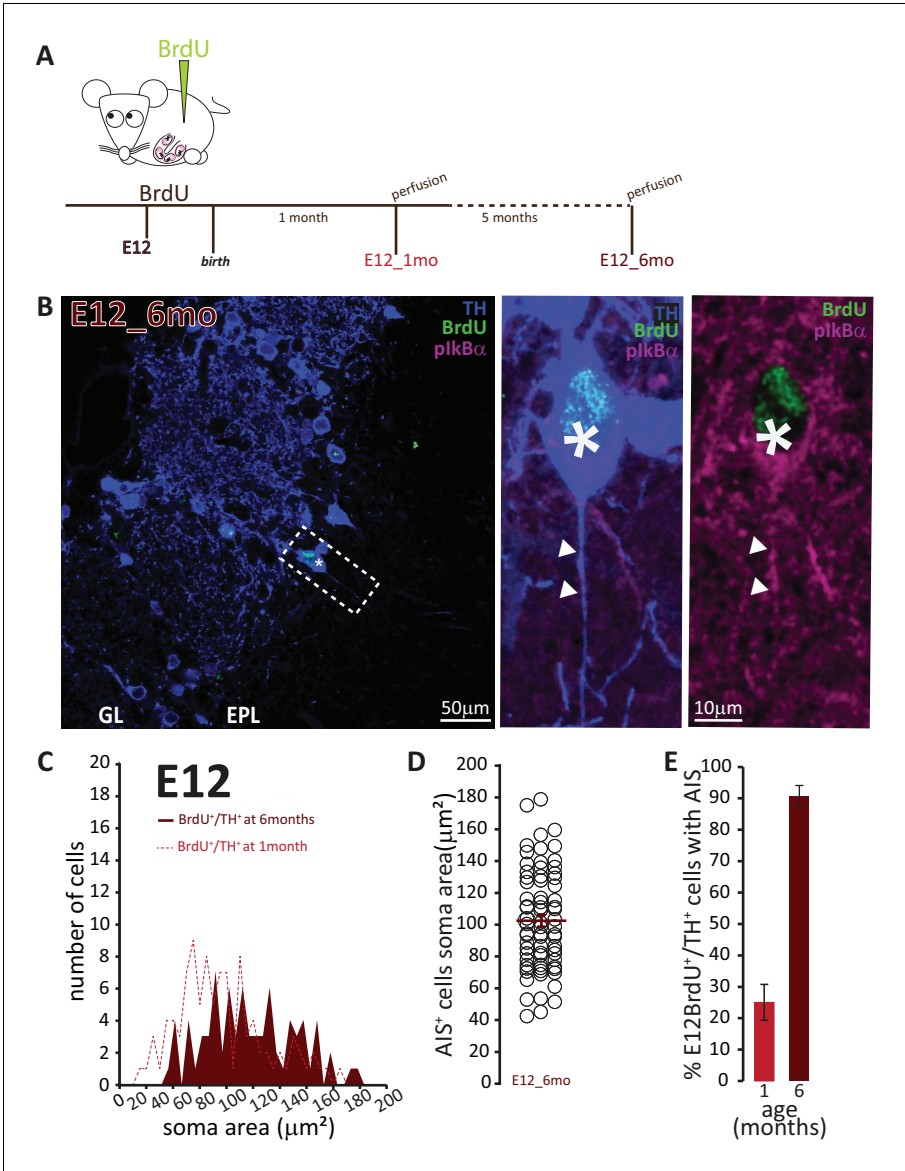

**Figure 7.** AIS-positive cells form the vast majority of embryonically-generated DA neurons that persist throughout adult life. (**A**) Schematic representation of the experimental strategy: pregnant wild-type mice were injected with a single dose of BrdU at E12. Tissue was collected from their offspring when they reached 6 months of age and compared with data collected from littermates perfused at 1 month of age (data shown in *Figure 6C*). (**B**) Left: Low-magnification example image of 6-month-old tissue stained with antibodies against BrdU (green), TH (blue) and the AIS marker pIκBα (magenta). GL, glomerular layer; EPL, external plexiform layer; asterisk indicates an E12-6mo BrdU+/TH+ DA cell; dashed line indicates the inset magnified on the right. Right: magnified example image of an E12-6mo BrdU+/TH+/pIκBα+cell; solid line indicates axon start; arrows show AIS start and end positions. (**C**) Soma area distribution of E12-6mo BrdU+/TH+ DA cells (dark red, n = 78, N = 4), overlaid on the soma area distribution of E12-1mo BrdU+/TH+ DA cells (dashed light red line; see *Figure 4C*). (**D**) Soma area of E12-6mo BrdU+/TH+/pIκBα+ DA cells. Empty circles represent individual AIS-positive neurons, dark red lines show mean ± SEM ($102 \pm 4 \ \mu m^2$; n = 71, N = 4). (**E**) Mean ± SEM percentage of AIS-positive E12-BrdU+/TH+ DA cells in tissue from 1-month-old (light red, 25%; n = 100, N = 3) and 6-month-old (dark red, 91%; n = 78, N = 4) mice.

DOI: https://doi.org/10.7554/eLife.32373.010

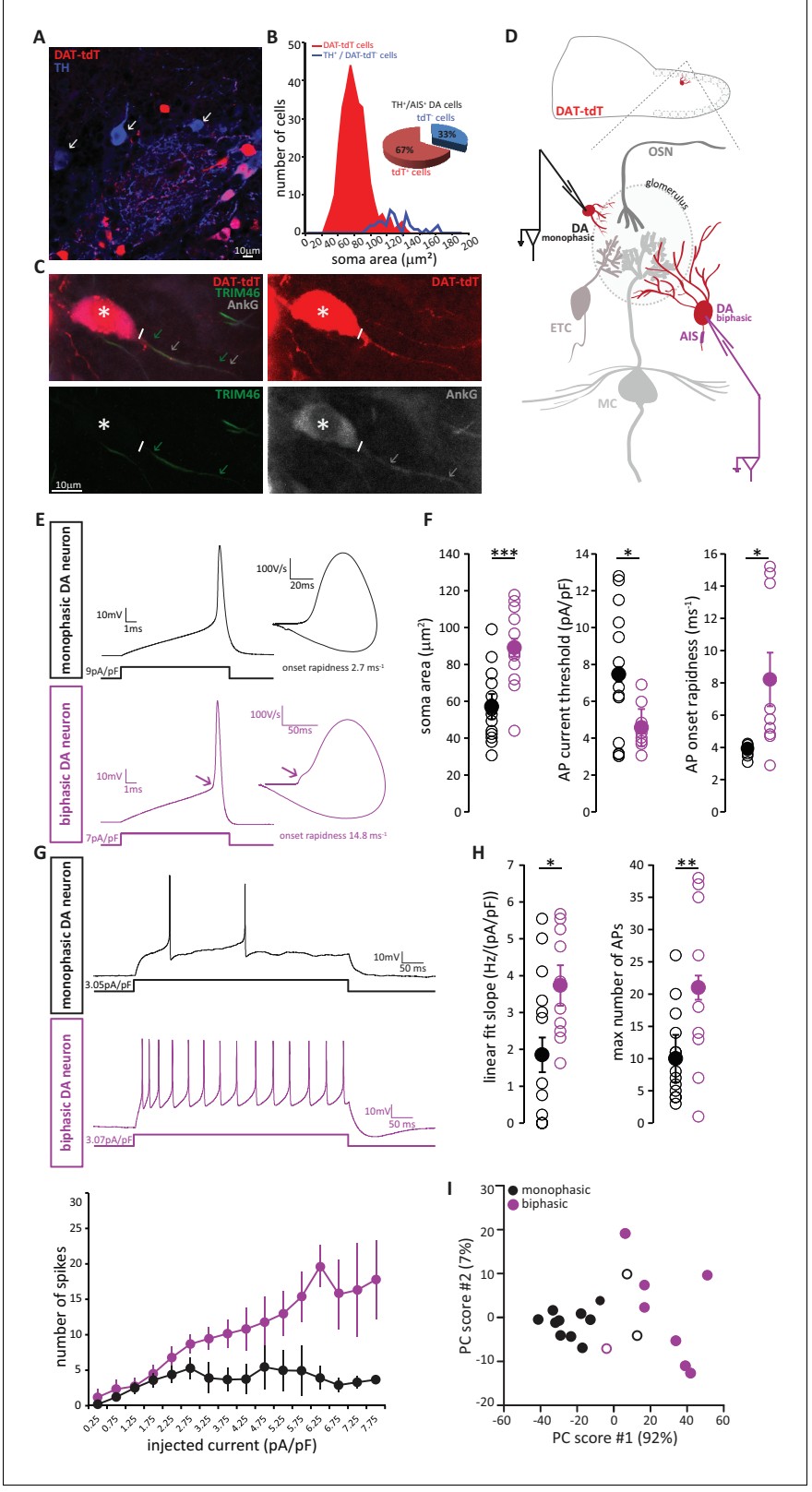

**Figure 8.** AIS-positive DA neurons have greater intrinsic excitability. (**A**) Example image of a fixed, 50 μm OB slice from a P28 DAT-tdT (red) mouse, immunostained with an anti-TH antibody (blue). While most TH-positive neurons exhibit red tdT fluorescence, some are tdT-negative (arrows). (**B**) Soma size distributions of all DAT-tdT-positive cells (red), and of DA neurons that are DAT-tdT-negative but TH-positive (blue). Inset: percentages of AIS-positive/

*Figure 8 continued on next page*

*Figure 8 continued*

TH-positive DA cells that are either tdT-positive or –negative (n = 50, N = 5). (**C**) Example image of a DAT-tdT-labelled DA cell (red) stained with the axonal marker TRIM-46 (green) and the AIS marker AnkG (greyscale). Asterisks indicate the soma; line indicates axon start; arrows indicate start and end position of the TRIM (green) and AnkG (white) label. (**D**) Schematic representation of the experimental strategy for whole-cell recordings: acute 300 µm OB slices were obtained from P21-35 DAT-tdT mice, and tdT-positive DA cells of either subtype were targeted for whole-cell patch-clamp recording. (**E**) Example current-clamp traces of single APs fired by monophasic (AIS-negative, black, n = 15) and biphasic (AIS-positive, magenta, n = 11) DAT-tdTomato neurons. Left: action potentials fired to threshold 10 ms somatic current injection. Right: phase plane plots of the spikes shown on the left. Arrow points to the AIS-dependent first action potential phase. (**F**) Quantification of soma area (t-test; ***p=0.0006), current threshold (Welch-corrected t-test; *p=0.017), and onset rapidness (Welch-corrected t-test; *p=0.035) in monophasic and biphasic cells. Empty circles show values from individual cells, filled circles show mean ± SEM. (**G**) Top: Example current-clamp traces of multiple APs fired in response to a 300pA/500 ms somatic current injection in monophasic and biphasic cells. Bottom: input-output curve of injected current density versus mean ± SEM spike number for each group. (**H**). Quantification of input-output slope (t-test; *p=0.044), and of the maximum number of action potentials fired by each cell over the whole range of injected current intensities (t-test; **p=0.0092). Empty circles show values from individual cells; filled circles show mean ± SEM. (**I**) Classification of DAT-tdT neurons based on values obtained from whole-cell recordings. Each circle shows one cell, plotted according to its primary and secondary PCA component scores (these components accounted for 92% and 7% of the variance in the data, respectively). Filled circles show cells correctly classified by k-means analysis; open circles show the few cells (3/26 overall) that were incorrectly classified.
DOI: https://doi.org/10.7554/eLife.32373.011

The following figure supplement is available for figure 8:

**Figure supplement 1.** AIS-positive DA neurons also have greater intrinsic excitability with strict, objective phase plane plot classification.
DOI: https://doi.org/10.7554/eLife.32373.012

AnkG revealed that only 67% of TH+/AIS+ DA neurons also expressed tdT (*Figure 8B,C*). However, tdT expression did not appear to reveal any further subdivision amongst the large, AIS-positive DA cells, because we found no difference in soma size between TH+/AIS+/tdT+ and TH+/AIS+/tdT- neurons (tdT+, mean ± SEM 118 ± 10 µm$^2$; tdT-, 126 ± 5 µm$^2$; Welch's corrected $t_{21.12}$ = 0.66, p=0.52). So, although DAT-tdT mice do not comprehensively reveal all bulbar DA neurons, visually targeting tdT-positive cells for electrophysiological recordings (*Figure 8C*) still enables functional comparisons to be made between AIS-positive and AIS-negative DA cell types.

Post-recording survival of bulbar DA cells for morphological or immunohistochemical analysis is notoriously difficult (A. Pignatelli, personal communication); this meant that we could not classify our recorded neurons as AIS-positive or AIS-negative on the basis of AnkG staining. Instead, we relied on a functional indicator of AIS presence: non-somatic action potential (AP) initiation. In phase-plane plots of single spikes fired in response to 10 ms somatic current injection, the site of AP generation can be inferred from the shape of the initial, rising component of the spike waveform (*Bean, 2007*; *Chand et al., 2015*; *Coombs et al., 1957*; *Jenerick, 1963*; *Khaliq et al., 2003*; *Shu et al., 2007*). While a smooth, monophasic phase plane plot is indicative of AP initiation at the somatic recording site, cells that initiate spikes at a distance from the electrode location – almost always at the AIS (*Bender and Trussell, 2012*; *Coombs et al., 1957*; *Foust et al., 2010*; *Kole et al., 2007*; *Palmer and Stuart, 2006*) - display a distinctive biphasic, or 'double-bumped' phase plane plot waveform (*Figure 8E*). We therefore divided our recorded DAT-tdT+ cells into monophasic and biphasic groups (see Materials and methods; *Figure 8—figure supplement 1*), which should be largely representative of AIS-negative and AIS-positive DA neurons, respectively. Indeed, we found that biphasic neurons were significantly larger than their monophasic counterparts (*Figure 8F*; *Table 2*; [*Chand et al., 2015*]).

We also identified several differences in intrinsic excitability between monophasic and biphasic DAT-tdT cells. Biphasic neurons generated single APs in response to lower amplitude somatic current injection, and initiated those APs more rapidly (*Figure 8F*; *Table 2*). When induced to fire repeatedly in response to longer lasting 500 ms somatic current injections of increasing intensity, biphasic cells displayed a linear input-output curve. Conversely, monophasic cells could not produce such a linear increase in spike number and soon reached a firing plateau (*Figure 8G*). This resulted

**Table 2.** Intrinsic electrophysiological properties of DAT-tdTomato neurons.

Mean values ± SEM of passive, action potential and repetitive firing properties for monophasic (putative AIS-negative, n = 15) and biphasic (putative AIS-positive, n = 11) DAT-tdTomato cells. Statistical differences between groups (monophasic vs biphasic) were calculated with a Student's t test for normally-distributed data ('t') or with a Mann–Whitney test for non-normally distributed data ('MW'). Bold type indicates statistically different measures, for which individual data points and example traces are presented in *Figure 5*.

**Intrinsic electrophysiological properties**

| | Monophasic (mean ± sem) | Biphasic (mean ± sem) | Test type, *p*-value |
|---|---|---|---|
| **Passive properties** | | | |
| Soma area (µm$^2$) | 57 ± 4.8 | 89 ± 6.8 | **t,<0.01** |
| Membrane capacitance (pF) | 19 ± 2 | 22 ± 2 | t, 0.39 |
| Resting membrane potential (mV) | −78 ± 1.9 | −74 ± 2.9 | MW, 0.31 |
| Input Resistance (MΩ) | 960 ± 272 | 572 ± 115 | MW, 0.13 |
| **Action potential properties** | | | |
| Threshold (pA/pF) | 7.5 ± 1.0 | 4.6 ± 0.4 | **t, 0.02** |
| Threshold (mV) | −30 ± 1.0 | −33 ± 1.0 | t, 0.13 |
| Max voltage reached (mV) | 19 ± 2.4 | 18 ± 4.0 | t, 0.80 |
| Peak amplitude (mV) | 49 ± 2.2 | 50 ± 4.4 | t, 0.79 |
| Width at half-height (ms) | 0.55 ± 0.03 | 0.50 ± 0.04 | t, 0.37 |
| Rate of rise (max dV/dt) (mV/ms) | 230 ± 16 | 251 ± 31 | t, 0.53 |
| Onset rapidness (1/ms) | 3.95 ± 0.28 | 8.22 ± 1.66 | **t, 0.03** |
| After hyper polarization AHP (mV) | −54 ± 1.4 | −55 ± 1.5 | t, 0.72 |
| AHP relative to threshold (mV) | 25 ± 1.3 | 24 ± 1.3 | t, 0.95 |
| **Repetitive firing properties** | | | |
| Rheobase (pA/pF) | 3.6 ± 1.0 | 1.7 ± 0.8 | MW, 0.28 |
| Max number of action potentials | 10 ± 2 | 21 ± 4 | **t, 0.01** |
| First action potential delay (ms) | 168 ± 38 | 273 ± 45 | t, 0.08 |
| Inter-spike interval CV | 0.28 ± 0.04 | 0.24 ± 0.03 | t, 0.46 |
| Slope of input/output curve (Hz/(pA/pF)) | 1.85 ± 0.54 | 3.53 ± 0.48 | **t, 0.04** |

DOI: https://doi.org/10.7554/eLife.32373.013

in monophasic cells having a significantly lower slope of their input-output curve, and a significantly lower maximum number of fired APs (*Figure 8H*; *Table 2*). While these differences in intrinsic excitability are certainly consistent with reported functional characteristics of AIS-positive versus AIS-negative neurons (*Chand et al., 2015*; *Zhou et al., 1998*; *Zonta et al., 2011*), we cannot rule out contributions from other, non-AIS-dependent factors (*Baranauskas et al., 2013*; *Eyal et al., 2014*; *Pignatelli et al., 2009*). Nevertheless, and regardless of their underlying cause, these physiological differences point to significantly greater intrinsic excitability in the biphasic, presumptive AIS-possessing DA subpopulation.

Finally, we asked whether the above measures from our whole-cell recordings could be reliably used to classify DAT-tdT neurons as belonging to either the biphasic/AIS-positive or the monophasic/AIS-negative subtype. Applying principal component analysis (PCA; see Materials and methods) to the five variables that differed significantly between mono- and biphasic DA cells generated primary and secondary component scores for each neuron that, when plotted against each other, revealed clear clustering by cell type (*Figure 8I*). Furthermore, using a k-means classification approach with the same data (see Materials and methods) we were able to assign our recorded cells to either the mono- or biphasic group with 85% accuracy. This suggests that, although there is considerable overlap in the functional properties of different subclasses of OB DA neuron, when taken together those properties reveal a significant distinction between putative AIS-positive and putative AIS-negative cell types.

## Large, putative AIS$^+$ DA neurons respond more weakly yet are more broadly tuned to odour stimuli

We next asked whether the morphological and physiological differences between the two subtypes of OB DA neuron are associated with distinct sensory response properties *in vivo*. Do different types of OB DA cell respond differently to olfactory stimuli?

To address this question, we employed a conditional mouse line in which the Cre-dependent Ca$^{2+}$ indicator GCamP6s was selectively expressed in OB DA neurons under the control of the DAT promoter (see Materials and methods). We could then characterise the sensory response properties of these cells by monitoring changes in GCaMP fluorescence while animals were presented with a panel of eight odour stimuli (see Materials and methods; *Figure 9A*; *Kapoor et al., 2016*). Given the current lack of a reliable *in vivo* live AIS marker, we classified DA neurons in these experiments using the proxy measure of soma size – this can be readily measured in live neurons, and is consistently associated with AIS-positive or AIS-negative identity across multiple datasets (*Figures 1*, *3* and *8*). Our upper bound for cells classed as 'small' was 70 µm$^2$, taken from the mean soma size of confirmed AIS-positive DA cells (*Figure 1B*, magenta distribution) minus two standard deviations (*i. e.* 136.7–2 × 33.2 µm$^2$). Under an assumption of normality, this cutoff excludes all but the smallest 2.5% of AIS-containing neurons. Similarly, our lower bound for cells classed as 'big' was 99 µm$^2$, taken from the mean soma area of confirmed AIS-negative DA neurons labelled via GFP electroporation at P1 (*Figure 4H*) plus two standard deviations (*i.e.* 65.6 + 2×16.6 µm$^2$). Again assuming normality, this cutoff excludes all but the biggest 2.5% of AIS-lacking cells.

We then analysed the odorant response properties of small/putative AIS-negative (n = 594) and big/putative AIS-positive (n = 622) GCaMP+ cells imaged in 13 mice. It immediately became apparent that different forms of odour-evoked responses could occur in these neurons. In many cases, a given odorant stimulus produced a relatively rapid increase in GCaMP fluorescence that then decayed back towards baseline – these responses, which we termed 'early excitatory' events, were the most prevalent form of odour-evoked signal in our DAT-GCaMP neurons (at least one early excitatory response was observed in 631/1216 = 52% of cells). In other cases, stimuli produced an increase in GCaMP intensity that had a delayed onset and peaked late in a given recording sweep – these 'late excitatory' events were readily and objectively distinguishable from early excitatory events (see Materials and methods; *Figure 9C,D*) and were less frequent in our sample (245/ 1216 = 20% of cells had at least one late excitatory response). Finally, we observed reasonably common examples of decreased GCaMP fluorescence upon odorant presentation. These 'inhibitory' events (at least one seen in 296/1216 = 24% of cells) usually had delayed onset, and were perhaps detectable because of the characteristically high spontaneous activity levels in OB DA neurons (*Chand et al., 2015*; *Pignatelli et al., 2005*; *Puopolo et al., 2005*). Although both late excitatory and inhibitory response types were unusual in their long peak latencies (but see similar long-latency excitatory responses in OB DA cells in *Banerjee et al., 2015*), their *Figure 2B*), and although they were more variable than early excitatory responses (*Figure 9E,F,G*), analysis of responses to individual stimulus presentations revealed them nevertheless to be reliable odour-evoked events. Peak latencies for all response types had coefficients of variation that were significantly less than one, indicative of non-random event timing across individual stimulus repeats (*Figure 9F*; early excitatory, mean ±SEM 0.28 ± 0.0063, Wilcoxon test vs 1, W = −198733, p<0.0001; late excitatory 0.40 ± 0.017, W = −29773, p<0.0001; inhibitory 0.38 ± 0.015, W = −42085, p<0.0001). In addition, no response type was generated by spurious, one-off fluctuations in fluorescence – mean amplitudes for the two stimulus repeats that produced the *weakest* responses were still significantly greater than one standard deviation above baseline (*Figure 9G*; early excitatory, mean ±SEM 5.38 ± 0.26, Wilcoxon test vs 1, W = 198036, p<0.0001; late excitatory 2.29 ± 0.20, W = 17963, p<0.0001; inhibitory 1.28 ± 0.060, W = 12954, p<0.0001). Crucially, neither of these measures of response reliability differed between small and big DA neurons, showing that late excitatory and inhibitory events were not only reliable per se, but were also just as reliable in both cell types (peak time CV, late excitatory, fixed effect of cell-type in multilevel ANOVA, F$_{1,232}$ = 1.89, p=0.17; inhibitory, F$_{1,289}$ = 0.12, p=0.75; mean amplitude of 2 weakest responses, late excitatory, F$_{1,212}$ = 0.83, p=0.36; inhibitory, F$_{1,177}$ = 0.19, p=0.67). Most response types occurred in isolation, although we did see some examples of combined excitatory-inhibitory responses (at least one seen in 76/1216 = 6% of cells).

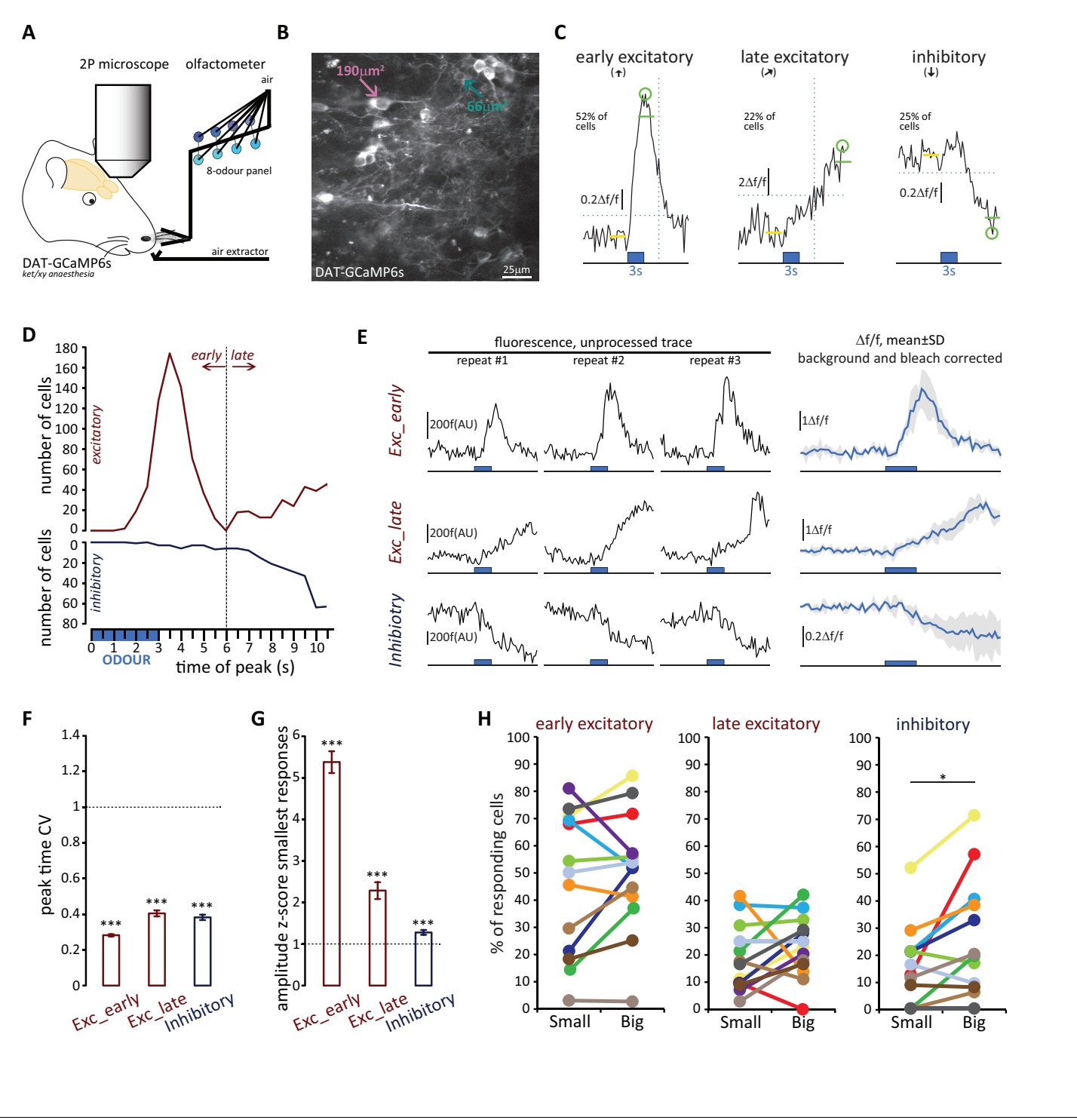

**Figure 9.** Both OB DA cell subtypes display diverse odour response types. (**A**) Schematic representation of the experimental strategy for *in vivo* recordings: adult DAT-GCaMP6s mice (anaesthetised with ketamine/xylazine) were presented with a panel of eight odours. Resulting changes in GCaMP fluorescence in DA neurons were imaged through a cranial window positioned over the OB. (**B**) Example field of view of the deeper part of the glomerular layer used for image acquisition (sum intensity projection of time axis, enhanced contrast). Fields of view were selected so as to contain both 'big' (soma area >99 µm², putatively AIS-positive; magenta arrow) and 'small' (soma area <70 µm², putatively AIS-negative; green arrow) DA neuron types. (**C**) Representative examples of the three categories of Δf/f GCaMP responses that we observed in the dataset. Deflections from baseline were considered events when they exceeded a threshold (horizontal dashed line) set at three times the baseline standard deviation. Excitatory responses occurred either quickly after odour presentation (early excitatory responses,[↑]; recorded in 52% of cells) or on a later timeframe (late

*Figure 9 continued on next page*

*Figure 9 continued*

excitatory responses, [↗]; recorded in 22% of cells). The early/late cut-off value was 6 s (vertical dashed line; see Materials and methods). Approximately a fifth of all cells also showed supra threshold negative deflection from baseline (inhibitory responses [↓]; present in 25% of cells). Green circle indicates absolute maximum or minimum value (peak) and green line shows the mean of the 3 s around the peak; yellow line indicates the 3 s of baseline prior to odour presentation (blue box). (D) Time of peak frequency plot for excitatory (red) and inhibitory (blue) events occurring after 3 s odour presentation (blue bar). The vertical dashed line indicates the cut-off value for early ($\leq$6 s) and late (>6 s) excitatory responses. (E) Left: Raw fluorescence traces of early-excitatory, late-excitatory and inhibitory responses to three repeated presentations of a 3 s odour stimulus (blue box). Right: Mean $\Delta f/f$ response over the three repeats, after background subtraction and bleach correction. Grey shading indicates standard deviation (SD). (F) Mean ±SEM coefficient of variation (CV) of peak time over the three odour presentations for early excitatory, late excitatory and inhibitory responses. Wilcoxon test vs. 1; ***p<0.001. (G) Mean ± SEM amplitude z-score value (mean of the two smallest responses out of the three odour repeats) for early excitatory, late excitatory and inhibitory responses. Wilcoxon test vs. 1; ***p<0.001. (H) Percentage of small (putative AIS-negative) and big (putative AIS-positive) cells in each mouse that showed at least one early excitatory (left), late excitatory (middle) or inhibitory (right) response. Colour-coding indicates the 13 mice that were imaged. Wilcoxon paired rank test; *p<0.05.

DOI: https://doi.org/10.7554/eLife.32373.014

Overall, 817/1216 = 67% of imaged DAT-GCaMP+ cells displayed at least one response type evoked by at least one of the eight odour stimuli we used.

All forms of odorant-evoked GCaMP response were observed in both big and small OB DA cell types. There were, however, some significant differences in their relative prevalence in the two neuronal populations. Paired, within-animal comparisons for the three major response types across the 13 mice in our sample revealed no significant differences in the proportions of small vs big DA neurons that displayed at least one odorant-evoked fast excitatory (*Figure 9H*; small cells, mean ±SEM 46 ± 7%; big cells 51 ± 6%; paired t-test, $t_{12}$ = 1.10, p=0.29), or slow excitatory (*Figure 9H*; small cells, 19 ± 3%; big cells 23 ± 3%; paired t-test, $t_{12}$ = 1.15, p=0.27) response. However, we did see a significantly higher proportion of inhibitory-responding neurons amongst the big cell population (*Figure 9H*; small cells, 16 ± 4%; big cells 26 ± 6%; Wilcoxon test, $W_{13}$ = 62, p=0.012).

To interrogate sensory stimulus selectivity further, we calculated a simple 'tuning index' (TI) for each response type in each cell, from the sum of all stimuli producing a significant change in GCaMP fluorescence (see Materials and methods). Cells with higher TI values responded to more odorants in our 8-stimulus panel. Although we acknowledge that this cannot represent a comprehensive description of tuning across all of odour space, this measure nevertheless allowed us to detect differences in response selectivity to a select group of odorant stimuli known to activate broad regions of the dorsal OB (*Livneh et al., 2014*; *Rokni et al., 2014*). In line with previous observations (*Banerjee et al., 2015*), we observed broad representations of odours in the responses of OB DA neurons (*Figure 10A*). The mean TI value for all excitatory responses (early + late combined) was 1.99 across all neurons in our sample, rising to 3.29 within the subset of neurons that displayed at least one excitatory response. Overall, this broad tuning was shared by both big and small OB DA sub-populations. However, we did observe significant cell-type-dependent differences in odour selectivity for particular response types. Importantly, we found only very weak correlations between TI measures calculated for the three major forms of odour-evoked response (early excitatory vs late excitatory, Spearman r = 0.018, p=0.53; early excitatory vs inhibitory, r = 0.13, p<0.0001; late excitatory vs inhibitory, r = 0.030, p=0.30; n = 1216 in all cases), suggesting that TI values for early excitatory, late excitatory and inhibitory events represent rather independent measures of tuning for distinct types of response produced by glomerular layer circuitry. To compare these TI measures between cell types, we needed powerful statistical tests that could leverage the large numbers of imaged neurons in our dataset whilst accounting for significant across-animal variability (see Materials and methods; *Figure 10B,C*). We therefore employed multilevel ANOVA analyses, where TI values from individual cells were compared between small versus big cell populations nested in animal subjects (see Materials and methods; [*Aarts et al., 2014*]). Using this approach, we found no effect of cell type on early excitatory TI values (*Figure 10B*; fixed effect of cell-type in multilevel ANOVA, $F_{1,1180}$ = 2.04, p=0.15). For the most prevalent form of odour-evoked response, then, tuning was strikingly similar in small and big OB dopaminergic neurons. For both late excitatory and inhibitory response types, though, the effect of cell type on TI was significant (*Figure 10B*; late excitatory, $F_{1,1214}$ = 5.58, p=0.018; inhibitory, $F_{1,1143}$ = 6.92, p=0.009), with big cells possessing consistently larger TI values on a mouse-by-mouse basis. When responding to odorant stimuli with late excitatory

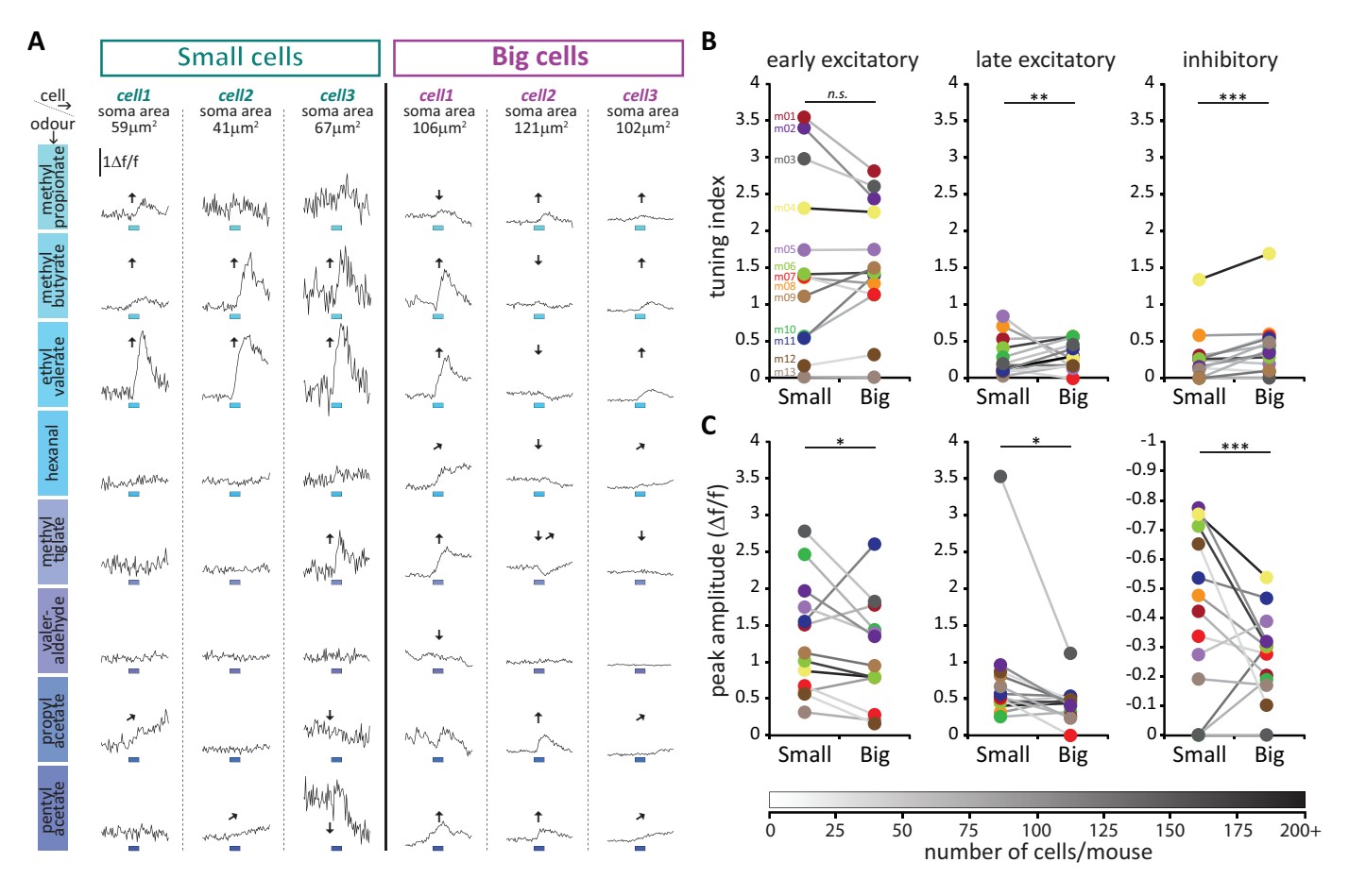

**Figure 10.** Large putative AIS-positive DA cells respond less strongly to odours and are more broadly tuned. (**A**) Example Δf/f GCaMP responses to the eight odours (rows, 3 s stimulus timing is indicated by blue shaded bars) for three big and three small example cells (columns; soma areas are indicated below the responses) imaged in the same mouse. Significant responses are indicated as: ↑ early excitatory, [↗] late excitatory, ↓ inhibitory. (**B**) Mean values of odour tuning indices for early excitatory (left), late excitatory (middle) and inhibitory responses (right) measured in small and big cells. Coloured dots indicate mean values for each cell type from each of the 13 imaged mice. The grayscale colour of the connecting lines indicates the number of recorded cells for each mouse (scale below). Cell-type effect in multilevel ANOVA; n.s., non-significant; *p<0.05; **p<0.01. (**C**) Similar to B. Median values of odour peak intensity for early excitatory (left), late excitatory (middle) and inhibitory responses (right). Cell-type effect in multilevel ANOVA; *p<0.05; ***p<0.0001.

DOI: https://doi.org/10.7554/eLife.32373.015

The following figure supplement is available for figure 10:

**Figure supplement 1.** Decay kinetics of late-latency responses do not differ between small and big OB DA neurons.
DOI: https://doi.org/10.7554/eLife.32373.016

or inhibitory events, therefore, big OB DA cells are significantly more broadly tuned than their small-soma neighbours.

Could this broader tuning in big OB DA neurons be explained by larger, more readily detectable odour-evoked responses in this cell type? Actually, measures of event amplitudes revealed the opposite to be the case: big cells had significantly weaker responses to odorant stimuli, a highly significant effect that held across all response types (*Figure 10C*; early excitatory, fixed effect of cell-type in multilevel ANOVA, $F_{1,626} = 6.58$, p=0.011; late excitatory, $F_{1,244} = 4.50$, p=0.035; inhibitory, $F_{1,295} = 32.69$, p<0.0001). Despite their higher intrinsic excitability (*Figure 8*), big, putative AIS+ DA neurons therefore do not display stronger responses to sensory stimuli *in vivo*. This unexpected effect may be because of fundamental differences between sensory stimulation *in vivo* versus direct electrical stimulation in vitro, or it may be due to cell-type differences in synaptic connectivity, or in the modulation of intrinsic properties in the intact OB. Additionally, it could be related to another

feature of *in vivo* GCaMP activity: baseline fluorescence. Resting fluorescence was significantly higher in big cells (fixed effect of cell-type in multilevel ANOVA, $F_{1,658}$ = 12.00, p=0.001), while baseline noise was significantly lower in this cell type ($F_{1,1211}$ = 41.84, p<0.0001), and both measures, especially noise, correlated strongly with all response amplitude measures (baseline fluorescence vs early excitatory amplitude, Spearman r = −0.49, p<0.0001, n = 612; vs late excitatory amplitude, r = −0.44, p<0.0001, n = 235; vs inhibitory amplitude, r = −0.58, p<0.0001, n = 282; baseline noise vs early excitatory amplitude, r = 0.67, p<0.0001, n = 631; vs late excitatory amplitude, r = 0.72, p<0.0001, n = 245; vs inhibitory amplitude, r = 0.83, p<0.0001, n = 296). The increased buffering capacity associated with higher resting GCaMP levels (*Svoboda et al., 1999*) could therefore lead to dampened response amplitudes in big cells. Additionally, lower spontaneous fluctuations in resting activity could allow big OB DA neurons to significantly respond to odorant stimuli with lower amplitude events. However, these cell-type distinctions in baseline activity cannot account for the differences in response selectivity between big and small cell populations (*Figure 10B*). Not only did we see identical big versus small cell selectivity for early excitatory events when baseline differences might be expected to influence tuning across all response types, we also observed only weak and inconsistent correlations with the different TI measures for both baseline fluorescence and noise (baseline fluorescence vs early excitatory TI, Spearman r = −0.05, p=0.07; vs late excitatory TI, r = 0.15, p<0.0001; vs inhibitory TI, r = 0.13, p<0.0001; n = 1164 in all cases; baseline noise vs early excitatory TI, r = 0.075, p=0.0085; vs late excitatory TI, r = −0.20, p<0.0001; vs inhibitory TI, r = −0.17, p<0.0001; n = 1216 in all cases). Finally, we wondered whether the prolonged timecourse and therefore truncated response profiles of late excitatory and inhibitory responses (*e.g. Figure 9C*, *Figure 10A*) might have contributed to the functional differences observed for these event types between big and small OB dopaminergic cells. However, accurate estimates of decay kinetics found no cell-type differences in either response type (see Materials and methods, *Figure 10—figure supplement 1*).

Overall in terms of odorant response properties, therefore, big/putative AIS-positive and small/putative AIS-negative OB DA neurons differ significantly in: 1) the broader selectivity of big cells for specific odour-evoked response types; and 2) the higher resting fluorescence, lower baseline noise, and smaller response amplitudes of big cells. Moreover, these two major functional features appear to be largely independent of each other.

## Discussion

Our results demonstrate the existence of two subtypes of OB DA neurons with distinct morphological, developmental and – crucially – functional characteristics. The majority of DA cells are small, locally projecting, anaxonic neurons which fire low numbers of somatic action potentials; they are continuously generated and turned over throughout life, and have stronger odorant-evoked responses that for certain event types are more narrowly tuned. Conversely, a minority of DA cells are large, wide-branching and equipped with an axon and an AIS, from which they generate high-frequency discharges of action potentials. These AIS-positive DA cells are born exclusively during early embryonic development and persist throughout life; their high excitability is nevertheless associated with weaker sensory-evoked responses in vivo, some types of which are more broadly tuned to odorant stimuli.

### Cell-type identity and functional diversity in neuronal circuits

An absolutely crucial step in understanding information processing in any neuronal network is to build an accurate classification of its component parts (*Zeng and Sanes, 2017*). Cell-type identity – as determined by ontology, gene expression, morphology, connectivity, and/or physiology – is intimately linked to the functional role that any neuron can play in a given circuit. It is therefore no surprise that in recent attempts to model realistic network operations, a great deal of effort has been spent delineating just how many component parts those networks contain. In different regions of the mammalian brain, we now have comprehensive descriptions of cell-type diversity with regards to, for instance, gene expression (*e.g. Romanov et al., 2017*; *Tasic et al., 2016*; *Zeisel et al., 2015*), neuronal morphology (*e.g. Cerminara et al., 2015*; *Parekh and Ascoli, 2015*), synaptic connectivity (*e.g. Morgan et al., 2016*), and sensory response properties (*e.g. Baden et al., 2016*), as well as combinatorial cellular-level identification schemes that multiplex several levels of description

(*e.g. Fuzik et al., 2016*; *Markram et al., 2015*; *Sanes and Masland, 2015*). These studies show that broad cell-type distinctions must be supplemented by fine-scale subdivisions within different cell types in order to fully understand network function. Such classification schemes are no less vital in our understanding of information processing in olfactory bulb circuits, where a uniquely modular topographic organisation of sensory inputs, coupled with the constant remodelling associated with both peripheral and central adult neurogenesis, promises novel insight into the way the brain interprets and adapts to the outside world.

However, our current understanding of functional diversity amongst neuronal populations in the olfactory bulb is far from complete. In glomerular layer circuits – the first networks to process sensory information arriving from the periphery – there is at least broad consensus on the division of juxtaglomerular neurons into excitatory and inhibitory cell types: glutamatergic, vGlut-expressing external tufted cells are, on the whole, readily distinguished from their GABA-positive interneuron neighbours (*Hayar et al., 2004*, but see *Tatti et al., 2014*). Furthermore, amongst those GABAergic interneurons are three neurochemically-distinct subpopulations, distinguishable (at least in mouse) by their non-overlapping expression of calretinin, calbindin, and tyrosine hydroxylase (*Kosaka and Kosaka, 2007*). However, although it has long been recognised that this latter group of TH-positive OB DA neurons are highly heterogeneous (*Davis and Macrides, 1983*; *Halász et al., 1981*; *Kosaka and Kosaka, 2007*; *Pignatelli et al., 2005*), there has been significant disagreement as to the precise nature of cell sub-type identity within this population. There is as yet no definitive classification of OB DA neurons, even though such a scheme is vital for our understanding of sensory processing functions in glomerular circuits.

Two major differing approaches to classifying OB DA neurons are currently under dispute. In the first, morphological considerations, especially the fact that many OB DA neurons spread their processes across more than one glomerulus (but see [*Bywalez et al., 2016*]), were used to label *all* of these neurons as superficial 'short-axon' cells (SACs; *Kiyokage et al., 2010*). This DA SAC population was then further subdivided into 'oligoglomerular' and 'polyglomerular' subtypes based on the extent of ramification across the glomerular layer (*Kiyokage et al., 2010*). In contrast, the second approach argues that classic morphological descriptions of superficial SACs report a complete lack of glomerular arborisation, and that the term 'SAC' should not be used to describe *any* OB DA neurons (*Kosaka and Kosaka, 2011*; *Kosaka and Kosaka, 2016*). Instead, according to this scheme, small-soma DA neurons form a subset of true periglomerular cells (DA-PGCs), while large-soma DA cells that project long distances across the glomerular layer are termed 'inhibitory juxtaglomerular association neurons', or IJGAs (*Kosaka and Kosaka, 2011*; *Kosaka and Kosaka, 2016*).

This lack of agreement has led to some studies simply grouping all DA neurons into a single neurochemically or genetically defined class (*e.g. Banerjee et al., 2015*). We agree that the dopaminergic-GABAergic phenotype of these cells is one of their most striking characteristics, defining them as a distinct population of OB interneurons. Moreover, we found here that on many measures the sensory response characteristics of the overall bulbar dopaminergic population are rather homogeneous (*Figures 9* and *10*). However, failing to identify important DA subclasses can produce issues in the interpretation of their functional roles within OB networks. The division we observe here may help to clarify matters substantially, and actually appears to fit reasonably well with both of the alternative schemes already proposed. On the one hand, AIS-positive, large OB DA neurons share many features with the 'polyglomerular' (*Kiyokage et al., 2010*) and 'IJGA' (*Kosaka and Kosaka, 2011*; *Kosaka and Kosaka, 2016*) classes. On the other hand, AIS-negative, small OB DA cells have much in common with the 'oligoglomerular' (*Kiyokage et al., 2010*) and 'DA-PGC' (*Kosaka and Kosaka, 2011*; *Kosaka and Kosaka, 2016*) subtypes. The AIS-negative class also shares important morphological features with a population of DAT-expressing 'clasping SACs' identified by recent live imaging of intracellular fills in acute OB slices (*Bywalez et al., 2016*), whose distinct dendritic architecture and predominantly juxtaglomerular arborisations would appear to separate them from classically-defined PGCs (*Kosaka and Kosaka, 2016*; *Pinching and Powell, 1971*). Most importantly, while soma size and dendritic spread are continuous variables that do not permit simple sub-group identification, the presence or absence of an axon is a discrete feature that should allow for cleaner classification. Indeed, segregating OB DA neurons based on axonal criteria has enabled important functional distinctions to be identified between subgroups (*Figure 8*; [*Chand et al., 2015*]) that were not evident from previous divisions based on continuous measures (*Pignatelli and Belluzzi, 2017*; *Pignatelli et al., 2005*). Finally, in terms of nomenclature, we certainly feel that 'SAC' is a misleading

term for all OB DA neurons, unless it is acknowledged that in some cells the axon in question is so short as to be non-existent. Perhaps, a simple distinction between 'axonic' and 'anaxonic' OB DA neurons will prove both clear and useful, although whether those subgroups represent forms of classically-defined PGC, SAC or other cell types can remain a matter for debate.

## Functional roles of axonic vs anaxonic DA neurons in sensory processing networks

The existence of two distinct subgroups of DA neurons raises the obvious question: how might these two subpopulations contribute to sensory processing? In the GL, inhibitory signalling can be either intraglomerular or interglomerular in nature – acting within the circuitry of an individual glomerulus, or acting between different glomeruli, respectively. Both AIS-positive and AIS-negative DA subtypes possess dendritic processes that ramify within the glomerular neuropil (*Figure 3*), so, assuming that the release of GABA and/or dopamine occurs from these dendrites in both cell types (*Borisovska et al., 2013*; *Kiyokage et al., 2017*; *Vaaga et al., 2017*, but see *Liberia et al., 2012*), both subpopulations have the potential to contribute to intraglomerular inhibition. This includes GABA and/or dopamine inhibiting release probability at OSN presynaptic terminals via the activation of $GABA_B$ and $D_2$ receptors, respectively (*Ennis et al., 2001*; *Hsia et al., 1999*; *Korshunov et al., 2017*; *McGann, 2013*; *Vaaga et al., 2017*). Local GABA release can also provide a brake on recurrent excitatory glomerular networks (*Gire and Schoppa, 2009*; *Murphy et al., 2005*; *Najac et al., 2011*), as well as effecting auto-disinhibition at high input strengths (*Parsa et al., 2015*). By acting at the levels of both input terminals and projection neuron dendrites, intraglomerular inhibition produced by both subtypes of OB DA neuron may subserve highly local gain control, potentially acting as a high-pass temporal and contrast filter to facilitate the detection of strong odorant stimuli (*e.g. Banerjee et al., 2015*; *Cavarretta et al., 2016*; *Cleland and Sethupathy, 2006*; *Gire and Schoppa, 2009*; *Korshunov et al., 2017*).

Interglomerular inhibition, by contrast, would appear to be restricted solely to the subpopulation of large, deep-lying, highly excitable AIS-positive OB DA neurons. Quite simply, only this population of GABAergic GL interneurons has an axonal process that can signal sufficient distances between glomerular networks (*Figures 1*, *2* and *3*; *Kiyokage et al., 2010*; *Kosaka and Kosaka, 2008*). Given the distribution of odorant information across glomeruli in a spatial map for odour identity (*Murthy, 2011*), such lateral inhibition between glomeruli has been predicted to enhance contrast between individual stimulus representations, and therefore aid odorant identification and/or discrimination (*e.g. Linster and Cleland, 2004*; *Uchida et al., 2000*; *Urban, 2002*). This lateral signal could be distributed by AIS-positive DA neurons via well-described long-range GABAergic monosynaptic connections onto external tufted cells (*Banerjee et al., 2015*; *Liu et al., 2013*; *Whitesell et al., 2013*). In addition, the distal interglomerular projections of (AIS-positive) OB DAs can induce rebound excitation via dopaminergic D1-receptor activation (*Liu et al., 2013*). Together with these cells' slightly more broadly-tuned late excitatory and delayed inhibitory responses to odorant stimuli (*Figure 10*), this delayed DA-mediated effect could contribute to the complex modulation of interglomerular dynamics, especially in the later stages of stimulus processing.

Finally, there might also be a significant developmental component to the relative functional contributions of axonic vs anaxonic OB DA neurons. Early in postnatal development, when neuronal activity contributes to the refinement of both OSN terminals (*Yu et al., 2004*; *Zou et al., 2004*) and projection neuron dendrites (*Lin et al., 2000*; *Matsutani and Yamamoto, 2000*) to individual glomeruli, the large, interglomerular-projecting AIS-positive DA cell type is relatively more numerous (*Figure 6B*). Maybe these inhibitory interneurons play a crucial role in co-ordinating odour-evoked and/or spontaneous activity across the glomerular layer at these early ages, allowing distinct activity patterns to drive anatomical segregation at the individual glomerulus level.

## A distinct role for functional plasticity in postnatally generated neurons?

Perhaps the most remarkable difference between the two axonic and anaxonic DA subtypes is that only the latter is generated throughout adult life. This observation is in agreement with a more general trend of adult-born neurons in the olfactory bulb, where neither PGCs nor granule cells possess an axon (*Lledo et al., 2006*). A recently observed small cohort of adult-born cortical neurons is also

anaxonic (*Le Magueresse et al., 2011*). In fact, with the notable exception of hippocampal dentate granule cells, it appears that all CNS neurons constitutively born during adulthood are anaxonic, contributing purely to local network activity by releasing neurotransmitter from their dendrites. Indeed, one may speculate that it is simpler for a newly generated neuron to insert itself in a pre-existing network without having to extend and connect a far-reaching axonal process. Accordingly, the large, axonic OB DA cells that do need to form such extensive connections are born only during early development at the same time that other projection neurons are populating the bulb (*Treloar et al., 2010*).

What is the evolutionary advantage for maintaining continuous neurogenesis of anaxonic local interneurons? The answer to this question is highly dependent on understanding the exact role of these small local neurons in olfactory processing. Immature adult-born neurons are distinguished by their heightened potential for activity-dependent plasticity (*Livneh and Mizrahi, 2012*). We might hypothesise, then, that a local intraglomerular gain control mechanism that can be readily modified by experience allows for broader behavioural flexibility, and permits rapid adaptation to new conditions in the external world (*Lazarini et al., 2014*; *Livneh and Mizrahi, 2012*; *Rochefort et al., 2002*). Moreover, if adult neurogenesis can be seen as an extreme form of structural plasticity that only one subtype of DA cells is capable of performing, it is probably reasonable to assume that more standard and less dramatic forms of plasticity are also differentially expressed by anaxonic and axonic cells. By definition, cells that do not have an AIS cannot undergo AIS plasticity, while *in vitro* evidence suggests that large axonic OB DA cells are capable of regulating the length and position of their AISs in an activity-dependent manner (*Chand et al., 2015*). But does this happen *in vivo*, and if so, does it have an impact on the cell's processing of olfactory inputs? Additionally, are other forms of activity-dependent plasticity in OB DA neurons (*Banerjee et al., 2013*; *Bonzano et al., 2016*; *Coppola, 2012*; *Hsia et al., 1999*; *Mizrahi, 2007*; *Wang et al., 2017*) specific to individual axonic versus anaxonic subclasses? Future studies will need to elucidate if other forms of neuronal plasticity can be induced in both DA cell subtypes in response to perturbations in sensory experience, and if so, how they impact on olfactory behaviour (*Taylor et al., 2009*; *Tillerson et al., 2006*).

# Materials and methods

## Key resources table

| Reagent type (species) or resource | Designation | Source or reference | Identifiers |
|---|---|---|---|
| Strain, strain background (*M. musculus*) | C57BL/6J mice | Charles River | Strain code 027 |
| Strain, strain background (*M. musculus*) | DAT-Cre, B6.SJL-Slc6a3tm1.1 (cre)Bkmn/J | The Jackson Laboratory | Jax stock 006660 |
| Strain, strain background (*M. musculus*) | VGAT-Cre, *Slc32a1*$^{tm2(cre)Lowl}$/J | The Jackson Laboratory | Jax stock 016962 |
| Strain, strain background (*M. musculus*) | flex-tdTomato, B6.Cg–Gt(ROSA) 26Sor$^{tm9(CAG-tdTomato)Hze}$, | The Jackson Laboratory | Jax stock 007909 |
| Strain, strain background (*M. musculus*) | flex-GCaMP6s animals, Ai96; B6;129S6-*Gt(ROSA) 26Sor*$^{tm96(CAG-GCaMP6s)Hze}$/J | The Jackson Laboratory | Jax stock 024106 |
| Transfected construct (Adeno-associated virus) | AAV9.EF1a.ChR2-YFP $^{lox/lox}$ virus | Penn Vector Core, USA | AV-9-PV1522, |
| Transfected construct (retrovirus) | floxed rv::dio-GFP$^{lox/lox}$ | Oscar Marin | *Ciceri et al., 2013* Nat Neuroscience 16(9):1199–210 |
| Antibody | polyclonal Anti-Tyrosine Hydroxylase, raised in Rabbit; use 1:500 | Millipore | catalogue number AB152; RRID:AB_390204 |
| Antibody | monoclonal Anti-Tyrosine Hydroxylase, raised in mouse; use 1:500 | Millipore | clone (LNC1) - catalog number MAB318; RRID: AB_2313764 |

*Continued on next page*

*Continued*

| Reagent type (species) or resource | Designation | Source or reference | Identifiers |
|---|---|---|---|
| Antibody | polyclonal Anti-Tyrosine Hydroxylase, raised in chicken; use 1:250 | Abcam | catalog number ab76442; RRID:AB_1524535 |
| Antibody | monoclonal anti-Ankyrin-G IgG2a, raised in mouse; use 1:500 | Neuromab | clone (106/36) - catalog number 75–146; RRID: AB_10673030 |
| Antibody | monoclonal anti-Ankyrin-G IgG2b, raised in mouse; use 1:500 | Neuromab | clone (106/65) - catalog number 75–147; RRID: AB_10675130 |
| Antibody | monoclonal anti-Ankyrin-G IgG1, raised in mouse; use 1:500 | Neuromab | clone (106/20) - catalog number 75–187; RRID:AB_10674433 |
| Antibody | monoclonal Phospho-IκBα (Ser32) (14D4), raised in rabbit; use 1:1000 | Cell Signaling Technology | catalog number 2859; RRID:AB_561111 |
| Antibody | polyclonal Anti-TRIM46, raised in rabbit; use 1:500 | Gift from Casper Hoogenraad | *van Beuningen et al. (2015)* Neuron. 88:1208–1226 |
| Antibody | monoclonal anti-MAP-2, raised in mouse; use 1:500 | Gift from Phillip Gordon-Weeks | |
| Antibody | monoclonal anti-chemical BrdU, raised in Rat; use 1:200 | Serotec | clone BU1/75 (ICR1)-catalog number OBT0030; RRID:AB_609568 |
| Antibody | polyclonal anti-GFP, raised in chicken; use 1:2000 | Abcam | catalog number ab13970; RRID:AB_300798 |
| Antibody | polyclonal anti-GFP, raised in guinea pig; use 1:500 | Synaptic Systems | catalog 132 005; RRID:AB_11042617 |
| Chemical compound, drug | Heparin | Alfa Aesar | CAS A16198 |
| C | | | |
| Chemical compound, drug | PIPES | Sigma | CAS P6757 |
| Chemical compound, drug | 5-Bromo-2′-deoxyuridine | Sigma | CAS 59143 |
| Chemical compound, drug | Alexa 488 | Thermo Fisher Scientific | A10436 |
| Chemical compound, drug | Methyl Propionate | Sigma | CAS 81988 |
| Chemical compound, drug | Methyl Butyrate | Sigma | CAS 246093 |
| Chemical compound, drug | Ethyl Valerate | Sigma | CAS 290866 |
| Chemical compound, drug | Hexanal | Sigma | CAS 115606 |
| Chemical compound, drug | Methyl Tiglate | Penta | CAS 13–73400 |
| Chemical compound, drug | Valeraldehyde | Sigma | CAS 110132 |
| Chemical compound, drug | Propyl Acetate | Tokyo Chemical Industry | CAS A0044 |
| Chemical compound, drug | Pentyl Acetate | Sigma | CAS 109549 |
| Chemical compound, drug | Diethyl Phthalate | Sigma | CAS 84662 |
| Software, algorithm | ImageJ software (Fiji) | NIH; *Schneider et al. (2012)* | RRID:SCR_003070 |

*Continued on next page*

*Continued*

| Reagent type (species) or resource | Designation | Source or reference | Identifiers |
|---|---|---|---|
| Software, algorithm | ClampFit 10.4 | pClamp | Molecular Devices; RRID:SCR_011323 |
| Software, algorithm | Prism 5.3 | GraphPad | RRID:SCR_002798 |
| Software, algorithm | Matlab | Mathworks | RRID:SCR_001622 |
| Software, algorithm | Vaa3D | Allen Institute for Brain Science | RRID:SCR_002609 |
| Software, algorithm | IBM SPSS Statistics | IBM | RRID:SCR_002865 |

## Animals

Unless otherwise stated, we used mice of either gender, and housed them under a 12 hr light-dark cycle in an environmentally controlled room with free access to water and food. Wild-type C57/Bl6 mice (Charles River) were used either as experimental animals, or to back-cross each generation of transgenic animals. The founders of our transgenic mouse lines – DAT-Cre (B6.SJL-$Slc6a3^{tm1.1(cre)Bkmn}$/J, Jax stock 006660), VGAT-Cre ($Slc32a1^{tm2(cre)Lowl}$/J, Jax stock 016962), flex-tdTomato (B6.Cg–Gt(ROSA)26Sor$^{tm9(CAG-tdTomato)Hze}$, Jax stock 007909), and flex-GCaMP6s animals (Ai96; B6;129S6-Gt(ROSA)26Sor$^{tm96(CAG-GCaMP6s)Hze}$/J, Jax stock 024106) – were purchased from Jackson Laboratories. If not stated otherwise, all experiments were performed at postnatal day (P) 28. All experiments were performed under the auspices of UK Home Office personal and project licences held by the authors, or were within institutional (Harvard University Institutional Animal Care and Use Committee) and USA national guidelines.

## Immunohistochemistry

Mice were anaesthetised with an overdose of pentobarbital and then perfused with 20 mL PBS with heparin (20 units.mL$^{-1}$), followed by 20 mL of 1% paraformaldehyde (PFA; in 3% sucrose, 60 mM PIPES, 25 mM HEPES, 5 mM EGTA, and 1 mM MgCl$_2$). The olfactory bulbs were dissected and post-fixed in 1% PFA for 2–7 d, then embedded in 5% agarose and sliced at 50 μm using a vibratome (VT1000S, Leica). Free-floating slices were washed with PBS and incubated in 5% normal goat serum (NGS) in PBS/Triton/azide (0.25% triton, 0.02% azide) for 2 hr at room temperature. They were then incubated in primary antibody solution (in PBS/Triton/azide) for 2 days at 4°C. The primary antibodies used and their respective concentrations are indicated in the key resources table. Slices were then washed three times for 5 min with PBS, before being incubated in secondary antibody solution (species-appropriate Life Technologies Alexa Fluor-conjugated; 1:1000 in PBS/Triton/azide) for 3 hr at room temperature. After washing in PBS, slices were incubated in 0.2% sudan black in 70% ethanol at room temperature for 3 min to minimise autofluorescence, and then mounted on glass slides (Menzel-Gläser) with MOWIOL-488 (Calbiochem). Unless stated otherwise, all reagents were purchased from Sigma.

## Birth-dating and sparse labelling

To birth-date neurons, we injected mice with a saline-based solution containing 50 mM bromodeoxyuridine (BrdU, Sigma) and 17.5 mM NaOH. Pregnant C57/BL6 female mice received one single intraperitoneal injection of this solution (0.075 ml/g) on the relevant gestational day; pregnancy start date (E0) was investigated twice daily and confirmed by the presence of a vaginal plug. Injected mothers and offspring were transcardially perfused on the relevant day, as detailed above. To permit BrdU detection, slices were first incubated in 2 M HCl for 30 min at 37°C, washed thoroughly and then processed for immunohistochemistry as described above.

Sparse morphological labelling was achieved by injecting 2 μl of AAV9.EF1a.ChR2-YFP $^{lox/lox}$ virus (AV-9-PV1522, Penn Vector Core, USA) in the lateral ventricle of P1-2 DAT$^{Cre}$ or VGAT$^{Cre}$ neonatal mice. A combination of birth-dating and sparse labelling was accomplished by either electroporating 2 μl of EGFP in the lateral ventricle of P1 C57/BL6 mice, or by injecting floxed rv::dio-GFP$^{lox/lox}$ retrovirus (*Ciceri et al., 2013*) in the lateral ventricle of E12 DAT$^{Cre}$ embryos. All invasive surgery was

performed under isoflurane anaesthesia, with Fast Green (0.3 mg/ml) co-injected to visually confirm positional accuracy.

Injections in embryos were performed with an injector and a 30.5 ga needle through the uterine wall into one of the lateral ventricles of the embryos. The uterine horns were then returned into the abdominal cavity, the wall and the skin were sutured, and embryos were allowed to continue their normal development.

Injections in neonates were performed in a semi-stereotaxic frame using a Hamilton syringe and a borosilicate glass capillary (GC100-15, Harvard Apparatus). For electroporation, after the injection of 2 µl of GFP in the lateral ventricle, five 50 ms-0.15 A electrical pulses were delivered at 1 Hz with plate electrodes (10 mm diameter, Nepagene, Japan) oriented in such a way to drive the current dorso-ventrally.

## Fixed-tissue imaging and analysis

All images were acquired with a laser scanning confocal microscope (Zeiss LSM 710) using appropriate excitation and emission filters, a pinhole of 1 AU and a 40x oil immersion objective. Laser power and gain were set to either prevent signal saturation in channels imaged for localisation analyses, or to permit clear delineation of neuronal processes in channels imaged for neurite identification (*e.g.* TH, GFP).

In *ex vivo* tissue, for branching patterns and reconstructions, images were taken with a 1x zoom (0.415 µm/pixel), 512 × 512 pixels, and in z-stacks with 1 µm steps. For AIS identification, images were taken with 3x zoom, 512 × 512 pixels (0.138 µm/pixel) and in z-stacks with 0.45 µm steps. All quantitative analyses were performed with Fiji (Image J). AISs were identified by confirming double labelling in 3D of TH along with a contiguous, elongated AnkG-positive stretch of neurite, and by then following the AnkG-positive process backwards to a clearly identifiable TH-positive cell body. Cell position in the glomerular layer (GL; defined by the perimeter of TH immunofluorescence) was classified as: a) 'upper' if the soma bordered with both the GL and the olfactory nerve layer; b) 'middle' if the soma was fully embedded in the GL; c) 'lower' if the soma bordered with both the GL and external plexiform layer (EPL); and d) 'EPL border' if the soma did not border at all with the GL. Soma area was measured at the cell's maximum diameter, by experimenters blind to cell type identity. Independent soma area measurements by two experimenters on a subset of TH-positive cells revealed extremely high levels of agreement (Pearson r = 0.97, n = 58). Soma-to-OSN layer distance was calculated from a straight perpendicular line connecting the cell body to the outer border of TH immunofluorescence. Morphological reconstructions of neurons that were sufficiently sparsely and brightly labelled were obtained with the auto-tracing Neuron 2.0 function in Vaa3D-3.20. Sholl analysis was performed on traced images using the automated Image J function, with a fixed first circle radius of 10 µm, and 5 µm increments for the following concentric circles. The number and length of primary dendrites was manually calculated with the freehand drawing function of Image J.

The density of AIS-positive DA neurons at P28 was calculated as follows. First, the average density of TH-positive cells per cubic millimetre of GL was calculated by manually counting cells and measuring GL area in medium-magnification stacks ($\sim$43,000 cells / mm$^3$; 40x objective, zoom 1; n = 14 slices, N = 3 mice). Second, the total volume of tissue used to identify a total of 297 AIS- and TH-positive cells was estimated by manually drawing GL area profiles in low-magnification images ($\sim$0.28 mm$^3$; 5x objective; n = 81, N = 9) and multiplying by the slice thickness (50 µm). The estimated number of TH-positive cells in this volume ($\sim$11,801) was then calculated by multiplying GL volume by the previously defined average TH-positive density per cubic millimetre. Finally, the AIS-positive/TH-positive cell percentage was calculated by dividing the total number of identified AIS-positive/TH-positive cells (297) by the estimated number of TH-positive cells present in the analysed volume. In P0 tissue, the proportion of axon-bearing DA neurons was calculated by manually counting the number of TH-positive cells in 20 stacks (n = 239, N = 2), and confirming with TRIM-46 co-label the subset of those with an axonal process (n = 14).

## Acute slice electrophysiology

P21-35 DAT$^{Cre}$-tdTomato mice were decapitated under isoflurane anaesthesia, and the OB was removed and transferred into ice-cold slicing medium containing (in mM): 240 sucrose, 5 KCl, 1.25 Na$_2$HPO$_4$, 2 MgSO$_4$, 1 CaCl$_2$, 26 NaHCO$_3$ and 10 D-Glucose, bubbled with 95%O$_2$ and 5% CO$_2$.

Horizontal slices (300 µm thick) of the olfactory bulb were cut using a vibratome (VT1000S, Leica) and maintained in ACSF containing (in mM): 124 NaCl, 2.5 KCl, 1.25 $Na_2HPO_4$, 2 $MgSO_4$, 2 $CaCl_2$, 26 $NaHCO_3$ and 15 D-Glucose, bubbled with 95% $O_2$ and 5% $CO_2$ for >1 hr before experiments began.

Whole-cell patch-clamp recordings were performed using an Axopatch amplifier 700B (Molecular Devices, San Jose, CA, USA) at physiologically relevant temperature (32–34°C) with an in-line heater (TC-344B, Warner Instruments). Signals were digitised (Digidata 1550, Molecular Devices) and Bessel-filtered at 3 kHz. Recordings were excluded if series (RS) or input (RI) resistances (assessed by −10 mV voltage steps following each test pulse, acquisition rate 20 KHz) were respectively bigger than 30 MΩ or smaller than 100 MΩ, or if they varied by >20% over the course of the experiment. Fast capacitance was compensated in the on-cell configuration and slow capacitance was compensated after rupture. Recording electrodes (GT100T-10, Harvard Apparatus) were pulled with a vertical puller (PC-10, Narishige) and filled with an intracellular solution containing (in mM): 124 K-Gluconate, 9 KCl, 10 KOH, 4 NaCl, 10 HEPES, 28.5 Sucrose, 4 $Na_2ATP$, 0.4 $Na_3GTP$ (pH 7.25–7.35; 290 MOsm) and Alexa 488 (1:150). DA cells were visualised using an upright microscope (FN1, Nikon, Tokyo, Japan) equipped with a 40X water immersion objective, and tdT/Alexa 488 fluorescence was revealed by LED (CoolLED pE-100) excitation. Post-patch fill with Alexa 488 was used both to confirm tdT-positive cell identity, and to measure soma area (ImageJ) in live images captured via a SciCam camera (Scientifica).

In current-clamp mode, evoked spikes were measured with $V_{hold}$ set to −60 ± 3 mV. For action potential waveform measures, we injected 10-ms-duration current steps of increasing amplitude until we reached the current threshold at which the neuron reliably fired an action potential ($V_m$ >0 mV; acquisition rate 200 KHz). For multiple spiking measures, we injected 500-ms-duration current steps from 0 pA of increasing amplitude (Δ2pA) until the neuron passed its maximum firing frequency (acquisition rate 50 KHz).

Exported traces were analysed using either ClampFit (pClamp10, Molecular Devices) or custom-written routines in MATLAB (Mathworks). Before differentiation for *dV/dt* and associated phase plane plot analyses, recordings at high temporal resolution (5 µs sample interval) were smoothed using a 20 point (100 µs) sliding filter. Monophasic versus biphasic phase plane plots were then visually determined independently by EG and MSG. We classified completely monotonic plots with continually increasing rate-of-rise as monophasic, and any plots showing a clear inflection in rate-of-rise over the initial rising phase as biphasic. Any discrepancies in classification were resolved by mutual agreement. We also corroborated our subjective classification using a quantitative measure of spike onset sharpness: the ratio of errors produced by linear and exponential fits to the perithreshold portion of the phase plane plot (*Baranauskas et al., 2010*; *Volgushev et al., 2008*). Fit error ratios were calculated with a custom Matlab script written by Maxim Volgushev, using variable initial portions of the phase plane plot between voltage threshold and 40% of maximum dV/dt (*Baranauskas et al., 2010*), for single spikes fired in response to 10 ms current injection at current threshold and up to three subsequent suprathreshold sweeps. Using strict, established (*Baranauskas et al., 2010*), but non-inclusive criteria for 'steep' (≈ biphasic; maximum fit error ratio >3) versus 'smooth' (≈ monophasic; maximum fit error ratio <1) spike onset, we were able to objectively classify phase plane plot shape in a smaller subset of our recorded DAT-tdT+ neurons. Of 13 neurons classified in this manner, only one was classified differently by subjective vs objective criteria. Importantly, the limited subset of objectively classified cells still displayed significant differences between mono- and biphasic OB dopaminergic neurons: as in the larger, subjectively-classified dataset, biphasic cells had bigger soma area, and were more excitable than their monophasic counterparts (*Figure 8—figure supplement 1*).

For quantification of AP properties, voltage threshold was taken as the potential at which *dV/dt* first passed 10 V/s. Onset rapidness was taken from the slope of a linear fit to the phase plane plot at voltage threshold. Spike width was measured at the midpoint between voltage threshold and maximum voltage. Rheobase and afterhyperpolarisation values were both measured from responses to 500 ms current injection, the latter from the local voltage minimum after the first spike fired at rheobase. Input-output curves were constructed by simply counting the number of spikes fired at each level of injected current density.

## *In vivo* imaging

Thirteen DAT<sup>Cre</sup>-GCaMP6s mice (either gender, age 4–10 months) were anaesthetised with a mixture of ketamine (100 mg/kg) and xylazine (10 mg/kg), placed in a stereotaxic apparatus and equipped with a cranial window over the olfactory bulbs using a sterile 3 mm biopsy punch (Integra Miltex). A custom-built titanium head plate was secured to their skull with adhesive luting cement (C and B Metabond, Parkell). A coverslip (3 mm, Warner Instruments) was placed over the cranial window and tissue adhesive (3M Vetbond) was used to secure the coverslip to the bone. The mice were allowed a minimum of a week to recover from surgery before the first imaging session. Prior to each imaging session the mice were newly anaesthetised with a mixture of ketamine (100 mg/kg) and xylazine (10 mg/kg) and secured in a custom-built microscope as described previously (*Kapoor et al., 2016*). Mice were given a maximum of one booster injection of anaesthesia per session, and were never imaged on consecutive days. GCaMP was excited and imaged via a water immersion objective (20x, 0.95 NA, Olympus; sterile saline was used as the fluid for the immersion objective) at 927 nm using a Ti:sapphire laser (Chameleon Ultra, Coherent) with 140 fs pulse width and 80 MHz repetition rate. Image acquisition, scanning, and stimulus delivery were controlled by custom-written software in LabVIEW (National Instruments). Eight odors were individually delivered via a custom-built olfactometer (*Kapoor et al., 2016*). The odour panel included: methyl propionate (Sigma, 81988), methyl butyrate (Sigma, 246093), ethyl valerate (Sigma, 290866), hexanal (Sigma, 115606), methyl tiglate (Penta, 13–73400), valeraldehyde (Sigma, 110132), propyl acetate (Tokyo Chemical Industry, A0044), and pentyl acetate (Sigma, 109549). All odours were diluted in diethyl phthalate solvent (Sigma-Aldrich, St. Louis, Missouri, United States) at 2% v/v.

Single-plane images of $300 \times 300$ pixel fields of view were acquired at 4 Hz during odour stimulation trials. Trial temporal structure consisted of: 7.5 s baseline, 3 s odour delivery, 7.5 s post-odour acquisition (18 s total, with 10 s inter-trial interval). All eight odours were probed sequentially, and then the entire block was repeated two more times, for a total of 24 odour trials for each field of view (three repetitions per odour). Cell soma selection was performed manually in Image J, using both the timecourse and the maximum intensity projections of each odour trial, and stored in a ROI mask for each field of view. Soma area and mean intensity values were then extracted for each ROI with a custom-written ImageJ macro, and saved as .xls and .txt files respectively. Data were then analysed with custom scripts in Matlab (Mathworks). Mean response traces were calculated across three individual stimulus presentations of each odour for each cell from background-subtracted raw fluorescence values, before bleach correction was carried out by extrapolation and subtraction of a single exponential function fitted to the 7.5 s pre-stimulus baseline. Mean bleach-corrected baseline fluorescence over a 3 s window immediately preceding odorant presentation (f) was then used to generate Δf/f traces. In some analyses, comparing baseline fluorescence values across different animals and imaging sessions (but not when calculating Δf/f values for all other measures), this baseline F value, averaged across all stimulus presentations, was normalised by the mean value for all small DA cells in a given field of view. The standard deviation of Δf/f values within the 3 s pre-stimulus period, averaged across all stimulus presentations, was taken as a measure of each cell's baseline noise. For each cell and each odorant, we then detected the point of maximum (for excitatory events) and minimum (for inhibitory events) Δf/f after stimulus onset, and took response amplitude as mean Δf/f over a 3 s window centred on this peak timepoint. Responses were classed as significant if this amplitude value was $\geq$3 x the baseline noise for that trace.

For analyses of response reliability (*Figure 9*), background subtraction, bleach correction and Δf/f calculation were carried out on individual response traces. Coefficients of variability for peak latencies were then calculated as sd/mean across three stimulus repeats. Z-scores were calculated on a repeat-by-repeat basis by dividing response amplitude (as above, mean Δf/f over a 3 s window centred on the peak timepoint) by the standard deviation of Δf/f values within the 3 s pre-stimulus period. We then calculated the mean of the two smallest amplitude z-scores across the three stimulus repeats.

Excitatory responses displayed a clear bimodal distribution of peak timepoints (see *Figure 9D*), so in an pre-analysed subsample of our data (n = 408 cells from N = 4 mice) we used unbiased k-means clustering on this parameter to set a threshold timepoint at 6 s after stimulus onset – all significant excitatory responses which peaked at or before this timepoint were classed as 'early', and all significant excitatory responses which peaked after this timepoint were classed as 'late'. Tuning

index (TI) values were calculated by summing the number of stimuli producing significant responses of a given type for each cell. These values were often zero, and many cells had zero TI values for all response types. These non-responding cells were included in all reported TI analyses, but results were identical in terms of significance if analyses were restricted only to those cells that displayed at least one significant response of any type to at least one odour. Amplitude measures were calculated for each cell as the mean across all odours that produced significant responses, but cell-type effects were also consistent if this was calculated as the maximum amplitude across all significant responses instead.

To estimate response decay constants for late excitatory and inhibitory responses, we started by identifying a subset of cells whose long-latency responses returned to baseline within the timeframe of our 18 s imaging sweep (late excitatory, n = 14; inhibitory, n = 44). In these cells, we compared the decay constants produced by a single exponential fit to either the 2.5 s immediately following the response peak (=10 timepoints at 4 Hz sample rate), or to the entire post-peak response profile. We found that fitting just the 2.5 s from the response peak produced accurate decay constant estimates – the differences between partial and full fits were small, and were not significantly different from zero for either response type (*Figure 10—figure supplement 1A,B*; late excitatory mean ±SEM - 3.35 ± 2.26 s, Wilcoxon test vs 0, W = −21, p=0.50; inhibitory, −0.41 ± 0.72 s, W = 221, p=0.17). Importantly, there was also no difference in the accuracy of this estimation in big versus small OB DA cells (late excitatory, fixed effect of cell type in multilevel ANOVA, $F_{1,9} = 0.92$, p=0.36; inhibitory, $F_{1,41} = 1.22$, p=0.28). We were then able to estimate decay constants in a larger subset of cells whose long-latency responses peaked at least 2.5 s before the end of our imaging sweeps. This subset comprised 42% and 44% of all cells with late excitatory and inhibitory responses, respectively, proportions that did not differ between cell types (late excitatory, small cells, 42/105; big cells, 60/140, Fisher's exact test, p=0.70; inhibitory, small cells 66/152, big cells 63/144, p>0.99). We found slow decay constants of approximately 8–11 s for both late excitatory and inhibitory response types, but no difference in decay kinetics between big and small OB DA cells (*Figure 10—figure supplement 1C,D*; late excitatory, fixed effect of cell type in multilevel ANOVA, $F_{1,98} = 0.32$, p=0.57; inhibitory, $F_{1,113} = 0.078$, p=0.78). Importantly, although making up only ~40% of the total population of responding cells, the subsample of neurons for which we could accurately estimate decay kinetics was representative of our sample as a whole, with no significant differences observed between decay-estimated and decay-non-estimated cells on a wide range of measures (t-test or Mann-Whitney as appropriate within small and big cell subpopulations, Bonferroni-corrected p>0.05 for soma area, baseline fluorescence, baseline noise, and response amplitude).

## Statistical analysis

Statistical analysis was carried out using Prism (Graphpad), Matlab (Mathworks) or SPSS (IBM). Sample distributions were assessed for normality with the D'Agostino and Pearson omnibus test, and parametric or non-parametric tests carried out accordingly. α values were set to 0.05, and all comparisons were two-tailed. Principal component analysis (PCA) and k-means classification on electrophysiological data were performed (Matlab functions 'pca.m' and 'kmeans_lpo.m', respectively) on the five variables that differed significantly between monophasic and biphasic DAT-tdT neurons. All were normally distributed except onset rapidness, which was rendered normal by logarithmic transform. Results of the k-means analysis were validated with a 'leave-one-out' protocol, which revealed cell-type classification to be robust to the removal of any one cell from the dataset. For multilevel analyses of in vivo GCaMP data, distributions of baseline noise and response amplitude measures were rendered normal by logarithmic transform, and outliers – defined as any value with an absolute z-score >3 – were removed (*Aarts et al., 2014*); a single outlier was removed from each dataset, representing <0.5% of each sample). These parameters were then analysed using linear mixed models (SPSS) with mouse as the subject variable. Tuning index data could not be rendered normal by any standard transforms, so were analysed using generalised linear mixed models with a negative binomial target distribution (accounting for >90% of sample variance; SPSS) and mouse as the subject variable. Dummy variable analysis revealed significant intracluster correlations in all cases, stressing the importance of nesting cell-by-cell data on individual mouse subjects (*Aarts et al., 2014*). Due to the non-normal nature of tuning index distributions, and the rarity of observing multiple different response types in any given single neuron, PCA was not attempted on our GCaMP data.

## Acknowledgements

This work was supported by a Sir Henry Wellcome fellowship (103044) to EG, a Wellcome Trust Career Development Fellowship (088301), BBSRC grant (BB/N014650/1) and ERC Consolidator Grant (725729; FUNCOPLAN) to MSG, a Medical Research Council 4 year PhD studentship to DJB, and an NIH grant (DC013329) to VNM. We wish to thank Casper Hoogenraad and Phillip Gordon-Weeks for antibodies, Maxim Volgushev for Matlab code, Mackenzie Mathis for DAT-GCaMP6s animals, Guilherme Neves and Lynette Lym for technical assistance with viral injections, and Vikrant Kapoor and Joseph Zak for guidance with in vivo imaging. Gordon Shepherd, Pierre-Marie Lledo, Oscar Marin, Juan Burrone, and all members of the Grubb and Murthy laboratories provided helpful discussions, while Alex Fleischmann and Ian Thompson made invaluable comments on the manuscript.

## Additional information

### Funding

| Funder | Grant reference number | Author |
| --- | --- | --- |
| Wellcome | 103044 | Elisa Galliano |
| Medical Research Council | MR/M501645/1 | Darren J Byrne |
| National Institutes of Health | DC013329 | Venkatesh N Murthy |
| European Research Council | 725729 FUNCOPLAN | Matthew S Grubb |
| Wellcome | 088301 | Matthew S Grubb |
| Biotechnology and Biological Sciences Research Council | BB/N014650/1 | Matthew S Grubb |

The funders had no role in study design, data collection and interpretation, or the decision to submit the work for publication.

### Author contributions

Elisa Galliano, Conceptualization, Formal analysis, Supervision, Funding acquisition, Investigation, Visualization, Writing—original draft, Project administration, Writing—review and editing; Eleonora Franzoni, Formal analysis; Marine Breton, Darren J Byrne, Formal analysis, Investigation; Annisa N Chand, Investigation; Venkatesh N Murthy, Conceptualization, Resources, Supervision, Funding acquisition, Project administration, Writing—review and editing; Matthew S Grubb, Conceptualization, Resources, Software, Formal analysis, Supervision, Funding acquisition, Writing—original draft, Project administration, Writing—review and editing

### Author ORCIDs

Elisa Galliano http://orcid.org/0000-0002-6941-766X
Venkatesh N Murthy http://orcid.org/0000-0003-2443-4252
Matthew S Grubb https://orcid.org/0000-0002-2673-274X

### Ethics

Animal experimentation: All experiments were performed under the auspices of UK Home Office personal and project licences held by the authors (Project Licenses: 70/7246 and 70/8906), or were within institutional (Harvard University Institutional Animal Care and Use Committee; Animal Protocol 29/20) and USA national guidelines.

### Decision letter and Author response

Decision letter https://doi.org/10.7554/eLife.32373.021
Author response https://doi.org/10.7554/eLife.32373.022

# Additional files

## Supplementary files
• Transparent reporting form
DOI: https://doi.org/10.7554/eLife.32373.017

## Data availability
The data is available at Dryad Digital Repository.

The following dataset was generated:

| Author(s) | Year | Dataset title | Dataset URL | Database, license, and accessibility information |
|---|---|---|---|---|
| Galliano E, Franzoni E, Breton M, Chand A, Byrne D, Murthy V, Grubb M | 2018 | Data from: Embryonic and postnatal neurogenesis produce functionally distinct subclasses of dopaminergic neuron | http://dx.doi.org/10.5061/dryad.b5hg8d6 | Available at Dryad Digital Repository under a CC0 Public Domain Dedication |

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
