## [Decision Letter]

Thank you for submitting your article "Embryonic and postnatal neurogenesis produce functionally distinct subclasses of dopaminergic neuron" for consideration by *eLife*. Your article has been reviewed by three peer reviewers, and the evaluation has been overseen by a Reviewing Editor and Eve Marder as the Senior Editor. The following individual involved in review of your submission has agreed to reveal his identity: Ottorino Belluzzi (Reviewer #2).

The Reviewing Editor has drafted this decision to help you prepare a revised submission.

Summary:

The manuscript by Galliano, Grubb and co-workers investigates dopaminergic interneurons in the olfactory bulb. Following up on previous work in cultured cells they show that this group of interneurons is actually made up of (at least) two, very different groups, AIS+ and AIS- cells. These differ in many features such as soma size, arborization, (somewhat obviously) excitability as well as – to a lesser and more overlapping degree – responsiveness in vivo. Importantly, AIS+ cells are exclusively generated during embryonic development, whereas AIS- cells are regenerated throughout the adult life. The paper is technically sound at the three levels of analyses (anatomical, slice electrophysiology and in vivo imaging). The results are important and may be useful in future studies.

Overall, the manuscript is well written and organized, however, there are several issues and questions that should be addressed so that a careful assessment of the contribution can be made.

Essential revisions:

1) "Small DA neurons are anaxonic" – the results and the interpretation of MAP2 as a "negative control" do not make any sense. The MAP2 data do not add any insight on top of the AnkG staining (and TRIM which overlaps with AnkG). The only argument for cells being anaxonic is that these cells do not have an AIS. The "negative control" would have been a complete lack of MAP2 in the axon of the large DA neurons. But since it is present in the axon and only "weakening at the AIS site", the use of MAP2 as negative evidence for an axon remains weak. In fact, the MAP2 results only weaken their argument because the difference between axons and dendrites in these particular neurons becomes now quite blurry.

2) "Large DA neurons have broader dendritic branching" – normally, dendritic tree measurements include numerous parameters but not necessarily "furthest branching point from the soma". Here it is the only measurement!

First, the authors should perform a serious morphometric analysis (they already have the data). Second, since the position of the soma of these neurons in the tissue is different, this measurement is perhaps the least informative of all because the dendrite could have longer distance to reach glomeruli by virtue of their position. Eyeballing the data, the great variability within groups makes the dendritic morphology a weak argument for identifying these specific types of neurons apart.

3) Large DA neurons are born earlier/small DA neurons are adult generated (Figure 4-5). Once the AIS has been established as a marker of the large soma DA neurons, this experiment is pretty much a repetition of the Kosaka and Kosaka paper from 2009 in Journal Neuroscience Research. Not much more.

4) Preponderance of large soma TH-PGNs at P0 and older age.

This part includes phrasing such as "…can already possess an AIS". This is making the data vague and fragmented. Why all the data are not presented? It would be helpful to show how many of the large TH-neurons do have an AIS and of those that have an AIS, how many are of specific size. The authors should add a table or Venn diagrams to show the numbers in full.

5) Large DA neurons are broadly tuned and weak responders

The classification of different response types in Figure 9C (subsection “Large, putative AIS+ DA neurons respond more weakly yet are more broadly tuned to odour stimuli”) is not convincing. Particularly the "late excitatory" and "inhibitory" seem a somewhat strange class of responses. The figure shows only a single trial. Please show more trials and the variance. We would assume that the further in time one goes away from stimulus onset, responses would be less uniform. We also suspect that three trials are too few for these strange delayed responses but as a first pass, they should at least show the data they collected.

The authors should present when the "late excitatory" and "inhibitory responses" return to baseline (the trace is cut at its peak). Otherwise it is hard to interpret these very slow responses. This classification was the basis of the main finding that differentiates between these two cell types. Given that these are rather strange responses (e.g. peaking ~6 sec after stimulus offset!), they should be clarified further.

The big and small cells have evidently different properties with regard to calcium measurements. This can be seen in the raw traces in 9D and the authors themselves relate to this issue (i.e. expression patterns, SNR, Calcium buffering by size or Calcium binding proteins, etc.). Thus, the naïve conclusions from these results is that calcium imaging may not be the best method to reveal differences between these particular cell types. Their responses beyond "technical" differences seem very similar.

6) The Materials and methods section that describes data (esp image) analysis should be expanded. How exactly is the GL defined in the images? How was the soma area measured (manually? Automatic? If manual how reliable is the manual measurement (e.g. are people measuring blind to other cellular features such as AnkG expression, what is the repeatability between measurements between people etc.)? (subsection “Fixed-tissue imaging and analysis”).

7) The biphasic / monophasic classification should be done more objectively. How good was the agreement / how many cells were disagreed upon? (subsection “Acute slice electrophysiology”, last paragraph).

8) The Discussion starts well but its second part "Functional roles…." lost real connection with the results of the paper. The attempt to use the data from this paper to sort out the mess in the field was not very convincing. Since the authors did not go the distance to study the contribution of either cell type to OB function it remains speculative. The authors should try to remain within the boundaries of the differences (or similarities) between the two cell types described in the paper. Unfortunately, these are not done convincingly.

For example, the differences between their cell types in anatomical reach (regardless of my reservations above) is about 100 microns or so (Figure 3). This translates into one glomerulus. So, the argument about "broadcasting" versus "glomerular specific" inhibition based on these differences is weak. The same goes for the "tuning broadness" but here the connection to the result is really far-fetched. Their data shows that these cell types differ by a tiny bit (Figure 9E – frankly, one can barely see the difference) and only for the specific responses (so called "late excitatory" and "late inhibitory"). Both response types are rare anyway (both <<1). Thus, calling these responses broad ("..their especially broad tuning for late…") is simply wrong. One cannot simply take these data and present as if one cell type is broadly tuned and the other is not. We suggest cutting considerably this part of the Discussion (subsection “Functional roles of axonic vs. anaxonic DA neurons in sensory processing networks”).

---

## [Author Response]

Essential revisions:1) "Small DA neurons are anaxonic" – the results and the interpretation of MAP2 as a "negative control" do not make any sense. The MAP2 data do not add any insight on top of the AnkG staining (and TRIM which overlaps with AnkG). The only argument for cells being anaxonic is that these cells do not have an AIS. The "negative control" would have been a complete lack of MAP2 in the axon of the large DA neurons. But since it is present in the axon and only "weakening at the AIS site", the use of MAP2 as negative evidence for an axon remains weak. In fact, the MAP2 results only weaken their argument because the difference between axons and dendrites in these particular neurons becomes now quite blurry.

We included the data in Figure 2 because we felt it was important not to make the assumption that the presence of an AnkG-positive segment necessarily indicated the presence of an axon. We therefore stained for MAP2 as a ‘dendritic marker’ that is renowned for its strong expression in the soma and throughout the full extent of all dendrites. Importantly, we found MAP2 localised along all processes in AIS-negative OB DA neurons, but saw that it was excluded from the post-AnkG axon in AIS-positive OB DA cells. We believe that, along with the use of TRIM46 as an AIS-independent marker of proximal axonal identity, these MAP2 data are instrumental in demonstrating that the AnkG-positive processes of large OB DA neurons are truly axonal.

However, we certainly recognise the reviewer’s concern over the seemingly aberrant localisation of MAP2 to the proximal axon in our AIS-positive neurons. In fact, although little noted, it is well documented that the proximal axon, especially the region preceding the AIS, does contain a significant concentration of MAP2 (van Beuningen et al., 2015; Gumy et al., 2017, from the latter: ‘the microtubule-associated protein 2 (MAP2), a well-known somatodendritic marker in multipolar neurons, was also detected in the initial part of the proximal axon but did not overlap with the AIS (Figures 1A and 1B)’). With the often distal location of the AIS in our OB DA neurons, the presence of MAP2 in the proximal axon is particularly clear (Figure 2C). Nevertheless, observing MAP2 localised to the pre-AIS region of the axon is not an indication that the process in question (which is TRIM46-positive and which completely lacks MAP2 from the AIS onwards) is any less ‘axonal’. We have therefore retained the MAP2 labelling shown in Figure 2, and for clarity on this issue have added a little more description and the above key references in the Results subsection “AIS-lacking DA neurons are anaxonic”.

2) "Large DA neurons have broader dendritic branching" – normally, dendritic tree measurements include numerous parameters but not necessarily "furthest branching point from the soma". Here it is the only measurement!First, the authors should perform a serious morphometric analysis (they already have the data). Second, since the position of the soma of these neurons in the tissue is different, this measurement is perhaps the least informative of all because the dendrite could have longer distance to reach glomeruli by virtue of their position. Eyeballing the data, the great variability within groups makes the dendritic morphology a weak argument for identifying these specific types of neurons apart.

We welcome the reviewer’s suggestion for additional morphological analysis, and now present these data in a revised Figure 3 and new Table 1. Quantitative analysis of Sholl intersection distributions revealed multiple significant differences between AIS-positive and AIS-negative OB DA neurons, all of which are indicative of broader dendritic branching in the AIS-positive subtype.

These additional data also directly address the reviewer’s second point that soma position might account for the cell-type difference in furthest intersection distance. Measuring the minimum distance of each labelled cell’s soma from the olfactory nerve layer confirmed that AIS-positive neurons sit on average ~67 µm deeper within the GL, which could potentially account for the same ~67 µm difference in furthest intersection difference between the cell types (Table 1). However, the difference in primary dendrite length between AIS-positive and AIS-negative neurons is only ~14 µm (Table 1), showing that the distance to first branch point and the onset of (presumably glomerular) arborisation cannot account for the full cell-type difference in furthest intersection values. More convincing, though, are the significant cell type differences in overall Sholl distributions (Figure 3E), area under curve, and maximum intersections (Table 1), all of which show that the dendrites of AIS-positive neurons are more widely branching rather than just being further displaced from the soma.

Finally, we completely agree on the ‘great variability within groups’ in terms of morphology and in fact explicitly noted this feature in the original manuscript. We have now stressed this point further with the undeniably ugly yet still significantly different Sholl distributions plotted in the new Figure 3E. Importantly, however, we do not attempt to use dendritic morphology to distinguish between OB DA subclasses – this would indeed be a ‘weak argument’. Instead, throughout the study we classify cells based on the binary presence or absence of an AIS (where possible, using reasonable proxies where not), but then investigate whether, on average, the different subgroups arising from that AIS-based classification are different on other morphological, developmental or functional characteristics.

3) Large DA neurons are born earlier/small DA neurons are adult generated (Figure 4-5). Once the AIS has been established as a marker of the large soma DA neurons, this experiment is pretty much a repetition of the Kosaka and Kosaka paper from 2009 in Journal Neuroscience Research. Not much more.

We recognise that the current study is built upon extensive foundations laid in this field by the Kosakas, and have appropriately and extensively cited their work throughout the manuscript, including the important 2009 paper noted by the reviewer. However, we believe the data presented in Figures 4 and 5 represent an important advance on the Kosaka and Kosaka (2009) study. Their paper showed a developmental distinction between OB DA subtypes based purely on soma size – a continuous variable which can be an indicative but not definitive descriptor of cell type identity. By contrast, the presence or absence of an AIS is not only a binary measure allowing the cleaner assignment of a cell into a given subgroup, but is also a morphological feature indicative of that cell’s axonic versus anaxonic nature (Figure 2), and by extension its potential output projections with OB networks (see Discussion). We therefore believe that our observations in Figures 4 and 5 constitute significantly more than a repetition of previous work.

4) Preponderance of large soma TH-PGNs at P0 and older age.This part includes phrasing such as "…can already possess an AIS". This is making the data vague and fragmented. Why all the data are not presented? It would be helpful to show how many of the large TH-neurons do have an AIS and of those that have an AIS, how many are of specific size. The authors should add a table or Venn diagrams to show the numbers in full.

The question of how many OB DA neurons are AIS-positive at different ages is indeed an important one. We have now included these data for P0 and P28 tissue in the main text (subsections “The axon initial segment is only present in a distinct subset of DA cells” and “AIS-positive DA neurons are exclusively born during early embryonic development, but anaxonic DA cells continue to undergo postnatal and adult neurogenesis”; see also additional Materials and methods subsection “Fixed-tissue imaging and analysis"), whilst stressing the important caveat that these values represent lower bound estimates because OB DA AISs are often difficult to find. The vague phrasing of our P0 observations has been replaced. We also note that the original, now unchanged Figures 1B and 6D showed the soma size distributions of AIS-positive cells at P28 and P0, respectively.

5) Large DA neurons are broadly tuned and weak respondersThe classification of different response types in Figure 9C (subsection “Large, putative AIS+ DA neurons respond more weakly yet are more broadly tuned to odour stimuli”) is not convincing. Particularly the "late excitatory" and "inhibitory" seem a somewhat strange class of responses. The figure shows only a single trial. Please show more trials and the variance. We would assume that the further in time one goes away from stimulus onset, responses would be less uniform. We also suspect that three trials are too few for these strange delayed responses but as a first pass, they should at least show the data they collected.The authors should present when the "late excitatory" and "inhibitory responses" return to baseline (the trace is cut at its peak). Otherwise it is hard to interpret these very slow responses. This classification was the basis of the main finding that differentiates between these two cell types. Given that these are rather strange responses (e.g. peaking ~6 sec after stimulus offset!), they should be clarified further.The big and small cells have evidently different properties with regard to calcium measurements. This can be seen in the raw traces in 9D and the authors themselves relate to this issue (i.e. expression patterns, SNR, Calcium buffering by size or Calcium binding proteins, etc.). Thus, the naïve conclusions from these results is that calcium imaging may not be the best method to reveal differences between these particular cell types. Their responses beyond "technical" differences seem very similar.

We entirely concur with these comments regarding the different types of odour response seen in our DAT-GCaMP cells, and welcome the opportunity to provide additional support for our observations. We have therefore performed extensive extra analyses of these data and added these results to the manuscript, as follows:

The late excitatory and inhibitory responses we observed are indeed ‘rather strange’ compared to the more commonly observed early excitatory events, although we do note that slow-latency excitatory responses have been reported (if not remarked upon) by others performing functional imaging in OB DA neurons (e.g. Banerjee et al., 2015, their Figure 2B). Given their unusual nature, we agree that our descriptions of the variability/uniformity/reliability of these responses needed to be much more detailed, so we now present data on precisely these questions in a dedicated new Figure 9. Here we show individual trials and mean ± sd traces for long-latency event types, as requested (Figure 9E). We also present two additional quantitative measures of response reliability: the CV of response peak times, and the normalised mean amplitude of the two smallest individual responses (Figure 9F, G; see Results subsection “Large, putative AIS+ DA neurons respond more weakly yet are more broadly tuned to odour stimuli”, and Materials and methods subsection “In vivo imaging”). On both measures it is clear that, as suggested by the reviewer, long-latency responses are indeed ‘less uniform’ than early excitatory events. However, the long-latency responses are nevertheless reliable phenomena per se: peak time CV is much less than 1 (Figure 9F), and the mean amplitude of the two weakest responses is still greater than 1 sd above baseline (Figure 9G). Moreover, both response types are equally reliable in both small and big DA neurons. We are now far more confident, therefore, that these long-latency events represent reliable, real and meaningful biological phenomena.

On the question of decay kinetics, comprehensively describing when the long-latency excitatory and inhibitory responses return to baseline was unfortunately impossible with our GCaMP imaging dataset – we were not expecting to observe such clear long-latency events, so our image acquisition was limited to 10.5 s after stimulus onset. However, we were able to estimate decay constants in a representative subset of our sample (>40% of cells; see Materials and methods subsection “In vivo imaging”) where post-peak decay could be reliably and accurately fitted with a single exponential function. These data are now presented in a new figure supplement (Figure 10—figure supplement 1) (also see Results subsection “Large, putative AIS+ DA neurons respond more weakly yet are more broadly tuned to odour stimuli”, sixth paragraph, Materials and methods subsection “In vivo imaging”, last paragraph). They show expectedly slow (τ ≈ 8-11 s) decay kinetics for both late excitatory and inhibitory events. Importantly, they reveal no differences in decay time constants for either response type between small and big DA neurons, suggesting that our premature truncation of these events, though regrettable, is unlikely to contribute to the cell-type differences we observed in odour response properties.

In the course of performing these additional analyses, we also identified data from 2 fields of view (n = 34 cells) in a single animal that were erroneously included in the originally presented GCaMP dataset despite clear motion artefacts. We apologise for this mistake, and have now excluded these data from the GCaMP analysis. All analyses and statistical tests have been re-run and all outputs have been changed where necessary in the subsection “Large, putative AIS+ DA neurons respond more weakly yet are more broadly tuned to odour stimuli”. In terms of significance, all results remain exactly the same as in the original submission.

On the suitability of calcium imaging for detecting functional differences between OB DA subtypes, we can only note that – as the reviewer says – we fully described and discussed the potential issues raised by baseline fluorescence, calcium buffering and SNR in the original manuscript. Despite these concerns, we were still able to detect some differences in odour response properties between big and small cells, and these differences appeared to be relatively independent of baseline Ca signal properties.

Finally, we completely agree with the reviewer’s reading that there are large similarities between the odour response characteristics of big and small OB DA neurons. For greater balance we have now altered the text to stress these similarities in both the Results (subsection “Large, putative AIS+ DA neurons respond more weakly yet are more broadly tuned to odour stimuli”, fourth paragraph) and Discussion (subsection “Cell-type identity and functional diversity in neuronal circuits”) sections.

6) The Materials and methods section that describes data (esp image) analysis should be expanded. How exactly is the GL defined in the images? How was the soma area measured (manually? Automatic? If manual how reliable is the manual measurement (e.g. are people measuring blind to other cellular features such as AnkG expression, what is the repeatability between measurements between people etc.)? (subsection “Fixed-tissue imaging and analysis”).

We have now expanded the Materials and methods section to address these issues in full (subsection “Fixed-tissue imaging and analysis”).

7) The biphasic / monophasic classification should be done more objectively. How good was the agreement / how many cells were disagreed upon? (subsection “Acute slice electrophysiology”, last paragraph).

A fully objective means of classifying phase plane plot shape would be an extremely welcome tool for the field, but we are not aware of any such method available at present. We were able, however, to use a published quantitative approach for classifying phase plane plot AP onset as either ‘steep’ (roughly indicative of a biphasic profile) or ‘smooth’ (≈ monophasic; Volgushev et al., 2009; Baranauskas et al., 2010; see Materials and methods subsection “Acute slice electrophysiology”, fourth paragraph). Using strict published criteria for these categorisations we were able to classify a subset of our patched DAT-tdT neurons (only a subset because these published criteria are non-inclusive, with many APs passing criteria for neither steep nor smooth onset), and this classification agreed very well with our subjective identification of mono- vs biphasic AP profiles (see Materials and methods aforementioned subsection). Crucially, even in this much-reduced subsample (total n = 13), we still saw that steep/biphasic/putative AIS-positive cells were larger and more excitable than smooth/monophasic/putative AIS-negative neurons. These data are now presented in a new figure supplement (Figure 8—figure supplement 1). We have also changed the Materials and methods text (subsection “Acute slice electrophysiology”, fourth paragraph) to correct an inaccuracy in our original description of the subjective classification process, for which we apologise – independent categorisation was indeed carried out by two experimenters, but discrepancies were then resolved by mutual consent rather than being discarded from the analysis.

8) The Discussion starts well but its second part "Functional roles…." lost real connection with the results of the paper. The attempt to use the data from this paper to sort out the mess in the field was not very convincing. Since the authors did not go the distance to study the contribution of either cell type to OB function it remains speculative. The authors should try to remain within the boundaries of the differences (or similarities) between the two cell types described in the paper. Unfortunately, these are not done convincingly.For example, the differences between their cell types in anatomical reach (regardless of my reservations above) is about 100 microns or so (Figure 3). This translates into one glomerulus. So, the argument about "broadcasting" versus "glomerular specific" inhibition based on these differences is weak. The same goes for the "tuning broadness" but here the connection to the result is really far-fetched. Their data shows that these cell types differ by a tiny bit (Figure 9E frankly, one can barely see the difference) and only for the specific responses (so called "late excitatory" and "late inhibitory"). Both response types are rare anyway (both <<1). Thus, calling these responses broad ("..their especially broad tuning for late…") is simply wrong. One cannot simply take these data and present as if one cell type is broadly tuned and the other is not. We suggest cutting considerably this part of the Discussion (subsection “Functional roles of axonic vs. anaxonic DA neurons in sensory processing networks”).

We have now considerably reduced this section of the Discussion as suggested. We have also removed all mention of broad vs. narrow tuning here, except for one brief mention related to late-latency, potentially dopaminergic interglomerular signalling (Liu et al., 2013), where we have been careful to refer to late-latency responses as ‘slightly more broadly tuned’. We have also altered the Abstract to remove all suggestion of overall broad tuning in large OB DA cells. However, because it is based purely on the binary morphological distinction between axonic and anaxonic DA cell types (Figure 2), we have retained in the ‘Functional roles’ section our discussion of the potentially specific contribution that AIS-positive DA cells can make to interglomerular inhibition.